# Tumor suppressor Tsc1 is a new Hsp90 co-chaperone that facilitates folding of kinase and non-kinase clients

Mark R Woodford[1,2,†], Rebecca A Sager[1,2,3,†], Elijah Marris[1,2,3], Diana M Dunn[1,2,3], Adam R Blanden[2,3], Ryan L Murphy[1,2], Nicholas Rensing[4,5], Oleg Shapiro[1,2], Barry Panaretou[6], Chrisostomos Prodromou[7], Stewart N Loh[2,3], David H Gutmann[4], Dimitra Bourboulia[1,2,3], Gennady Bratslavsky[1,2], Michael Wong[4,5] & Mehdi Mollapour[1,2,3,*]

## Abstract

**The tumor suppressors Tsc1 and Tsc2 form the tuberous sclerosis complex (TSC), a regulator of mTOR activity. Tsc1 stabilizes Tsc2; however, the precise mechanism involved remains elusive. The molecular chaperone heat-shock protein 90 (Hsp90) is an essential component of the cellular homeostatic machinery in eukaryotes. Here, we show that Tsc1 is a new co-chaperone for Hsp90 that inhibits its ATPase activity. The C-terminal domain of Tsc1 (998–1,164 aa) forms a homodimer and binds to both protomers of the Hsp90 middle domain. This ensures inhibition of both subunits of the Hsp90 dimer and prevents the activating co-chaperone Aha1 from binding the middle domain of Hsp90. Conversely, phosphorylation of Aha1-Y223 increases its affinity for Hsp90 and displaces Tsc1, thereby providing a mechanism for equilibrium between binding of these two co-chaperones to Hsp90. Our findings establish an active role for Tsc1 as a facilitator of Hsp90-mediated folding of kinase and non-kinase clients— including Tsc2—thereby preventing their ubiquitination and proteasomal degradation.**

**Keywords** Aha1; heat-shock protein 90; Tsc1; Tsc2; tuberous sclerosis complex

**Subject Categories** Protein Biosynthesis & Quality Control; Signal Transduction

The EMBO Journal (2017) 36: 3650–3665

## Introduction

Molecular chaperones, as indicated by their names, are involved in folding, stability, and activity of many proteins also known as "client proteins" (Schopf *et al*, 2017). The molecular chaperone heat-shock protein 90 (Hsp90) looks after many clients (> 200 known) that are involved in tumor initiation, progression, and metastasis. Therefore, Hsp90 is recognized as a suitable target for cancer therapy (Neckers & Workman, 2012). Hsp90 chaperone function is linked to its ability to bind and hydrolyze ATP and depends on a cycle of Hsp90 conformational changes (Prodromou, 2012). Hsp90 drugs bind to the ATP-binding pocket of Hsp90 and inhibit its chaperone activity. In turn, this prevents Hsp90 association with client proteins, causing their proteasomal degradation. Unlike other anti-cancer drugs, Hsp90 inhibitors target multiple drivers of oncogenesis simultaneously.

Co-chaperones regulate Hsp90 chaperone cycle by directly interacting with Hsp90 and providing directionality to the cycle (Cox & Johnson, 2011). Additionally, certain co-chaperones, for example, HOP and Cdc37[p50], decelerate or inhibit the Hsp90 chaperone cycle, loading distinct sets of clients such as steroid hormone receptors and kinases, respectively, to the Hsp90 (Verba *et al*, 2016). Conversely, the Aha1 co-chaperone aids in high-energy conformational modulations necessary for Hsp90 ATPase competence. This markedly increases the weak endogenous enzymatic activity of Hsp90 and establishes Aha1 as a crucial component of active Hsp90 chaperone complexes (Panaretou *et al*, 2002; Li *et al*, 2013a).

Tuberous sclerosis complex (TSC) is a rare autosomal dominant syndrome that is characterized by the development of benign tumors in various organs, including skin, brain, heart, kidneys, and

1 Department of Urology, SUNY Upstate Medical University, Syracuse, NY, USA
2 Upstate Cancer Center, SUNY Upstate Medical University, Syracuse, NY, USA
3 Department of Biochemistry and Molecular Biology, SUNY Upstate Medical University, Syracuse, NY, USA
4 Department of Neurology, Washington University School of Medicine, St. Louis, MO, USA
5 Hope Center for Neurological Disorders, Washington University School of Medicine, St. Louis, MO, USA
6 Institute of Pharmaceutical Science, King's College London, London, UK
7 Genome Damage and Stability Centre, University of Sussex, Brighton, UK
*Corresponding author. Tel: +1 315 464 8749; E-mail: mollapom@upstate.edu
†These authors contributed equally to this work

bladder (Henske *et al*, 2016). TSC is also frequently associated with epilepsy, intellectual disability, and autism (Banerjee *et al*, 2011; Zeng *et al*, 2011). Mutations in one of the tumor suppressor genes, *TSC1* or *TSC2*, cause TSC (Huang & Manning, 2008). Tsc2 protein (tuberin) of approximately 200 kDa in size behaves as a GTPase-activating protein (GAP), and Tsc1 (hamartin) with a molecular weight of 130 kDa is required to stabilize Tsc2 and prevent its ubiquitin-mediated degradation (Benvenuto *et al*, 2000). More specifically, Tsc1 prevents the interaction between Tsc2 and the E3 ubiquitin ligase HERC1 (Chong-Kopera *et al*, 2006). However, the precise protective role of Tsc1 toward Tsc2 remains elusive. Here, we show Tsc1 is a new co-chaperone of Hsp90. Tsc1 carboxy-domain forms a homodimer and it binds to the Hsp90 middle domain where it inhibits Hsp90 ATPase activity and also blocks interaction of the activating co-chaperone Aha1 with the Hsp90 middle domain. Tsc1 also increases the binding of Hsp90 to ATP and inhibitors. Tsc1 has a higher affinity for binding to Hsp90 than Aha1. However, this affinity is reversed when c-Abl-mediated phosphorylation of Aha1-Y223 increases its binding to Hsp90 and displaces Tsc1. This provides a mechanism of equilibrium between these two co-chaperones binding to Hsp90. Tsc1 functions as a facilitator of Hsp90 in chaperoning the kinase and non-kinase clients including Tsc2, therefore preventing their ubiquitination and degradation in the proteasome.

# Results

## Tsc2 is a client of Hsp70 and Hsp90

Molecular chaperones are generally involved in the folding and stability of proteins. We therefore tested whether the integrity of Tsc2 depended on the Hsp70 and Hsp90 chaperones. Endogenous Tsc2 was immunoprecipitated from HEK293 cells and it was shown to interact with both Hsp70 and Hsp90 and their co-chaperones including HOP, Cdc37[p50], PP5, and p23, but not Aha1 (Figs 1A and EV1). We next showed that treating HEK293 cells with the Hsp70 inhibitor JG-98 (Fig 1B) (Li *et al*, 2013b), or Hsp90 inhibitors ganetespib GB (Fig 1C) (Ying *et al*, 2012), or SNX2112 (Fig 1D) (Barrott *et al*, 2013) cause the degradation of Tsc2. Surprisingly, inhibition of Hsp70 or Hsp90 did not affect the stability of Tsc1 (Fig 1B–D). It is noteworthy that we used the degradation of Akt and phospho-S473-Akt as a positive control for Hsp90 inhibition in cells, since Akt is a *bona fide* Hsp90 client protein.

Inhibition of Hsp70 or Hsp90 generally leads to ubiquitination and degradation of its client proteins in the proteasome (Xu *et al*, 1999). Treating HEK293 cells with 50 nM proteasome inhibitor bortezomib (BZ) for 2 h stabilizes Tsc2 (Fig 1E). These data suggest that Tsc2 experiences protein turnover at steady state. We next showed that treating HEK293 cells with BZ for 1 h prior to addition of GB (Hsp90 inhibitor) blocked the degradation of Tsc2 (Fig 1F). Finally, to determine whether Hsp70 or Hsp90 inhibition leads to ubiquitination of Tsc2 prior to its degradation in the proteasome, Tsc2-FLAG was transiently expressed, immunoprecipitated, and salt-stripped (with 0.5 M NaCl) from HEK293 cells treated with either 50 nM BZ, 1 μM GB, or 10 μM JG-98 for 4 h. Tsc2 is ubiquitinated upon inhibition of Hsp70, Hsp90, or the proteasome (Fig 1G). Taken together, Tsc2 appears to be a new client of Hsp70 and

Hsp90, and pharmacologic inhibition of Hsp90 leads to ubiquitination and degradation of Tsc2 in the proteasome.

## The new co-chaperone Tsc1 inhibits Hsp90 ATPase activity

Tsc1 stability does not depend on Hsp70 and Hsp90; however, because of its relationship with Tsc2, we asked whether Tsc1 interacts with these two molecular chaperone machineries. We first immunoprecipitated endogenous Tsc1 from HEK293 cells and detected both Hsp70 and Hsp90 (Fig 2A). We also observed Tsc1 interaction with Hsp90 co-chaperones PP5 and Cdc37[p50] (Fig 2A) but not Aha1, HOP, or p23 (Fig EV2A). We also co-immunoprecipitated Hsp90 client proteins Raf-1, Akt, Cdk4, glucocorticoid receptor (GR), and Tsc2 (Fig 2A). Hsp90 consists of N (amino)-, M (middle)-, and C (carboxy)-domains. To determine Tsc1 and Tsc2 interaction with these Hsp90 domains, we transiently transfected and expressed each domain with FLAG-tag in HEK293 cells. Using anti-FLAG M2 affinity gel, each Hsp90 domain was immunoprecipitated and co-immunoprecipitation of Tsc1 and Tsc2 was observed by immunoblotting. Tsc1 interacts with the Hsp90α M- and C-domains (Fig 2B), whereas Tsc2 bound only to the Hsp90α C-domain (Fig 2B). The MEEVD motif at the extreme C-terminus of Hsp90 is a highly conserved tetratricopeptide repeat (TPR) domain-binding site, which mediates interaction with many co-chaperones (Chadli *et al*, 2008). Deletion of MEEVD from Hsp90α (αΔTPR) or Hsp90β (βΔTPR) completely abrogated the interaction of Hsp90 with Tsc1 (Fig 2C).

We next determined the region in Tsc1 that interacts with Hsp90. Based on previous work, we decided to express FLAG-tag Tsc1-fragment-A (N-domain; amino acid 1–301), Tsc1-fragment-B (Tsc2-binding domain; amino acid 302–430), and Tsc1-fragment-D (C-domain; amino acid 998–1,164) in HEK293 cells. Our data indicate that the carboxy-domain of Tsc1 (amino acid 998–1,164 or fragment D) interacts with Hsp90α (Fig 2D). In fact, Tsc1-D-HA interacts with the middle domain of Hsp90α-FLAG (Fig 2E). It is noteworthy that we were unable to express Tsc1-fragment-C (amino acid 431–718) in HEK293 cells.

We next bacterially expressed and purified fragment D (Tsc1-D-His$_6$) and examined its affinity for binding to Hsp90α in the presence of ADP, AMPPNP, and GB *in vitro* by fluorescently labeling Tsc1-D-His$_6$ with Texas Red maleimide, and measuring the $K_d$ by fluorescence anisotropy (Fig 2F). Our bacterially expressed and purified Hsp90α was functional based on its ATPase activity. The titration fit to a single-site binding equation with a $K_d$ of $0.48 \pm 0.19$ μM (Fig 2F). This $K_d$ was not significantly affected by the presence of ADP or non-hydrolyzing AMPPNP (Fig 2F). However, the presence of GB produced a $K_d$ of $> 5$ μM, therefore suggesting Tsc1-D-His$_6$ binding to Hsp90 is significantly reduced in the presence of GB. Binding of bovine serum albumin (BSA) was used as a negative control (Fig EV2B).

The chaperone function of Hsp90 is linked to its ATPase activity (Panaretou *et al*, 1998). We therefore tested the impact of Tsc1 on Hsp90 ATPase activity. Recombinant Hsp90α and Tsc1-D-His$_6$ were used in the molar ratio indicated (Figs 2G and EV2C and D) in the PiPer Phosphate Assay Kit (Thermo Fisher Scientific), in the presence of ATP as substrate. Hsp90α ATPase activity was measured *in vitro* as previously described (see Materials and Methods; Dunn *et al*, 2015; Figs 2G and EV2C and D). 10 μM GB inhibited Hsp90α

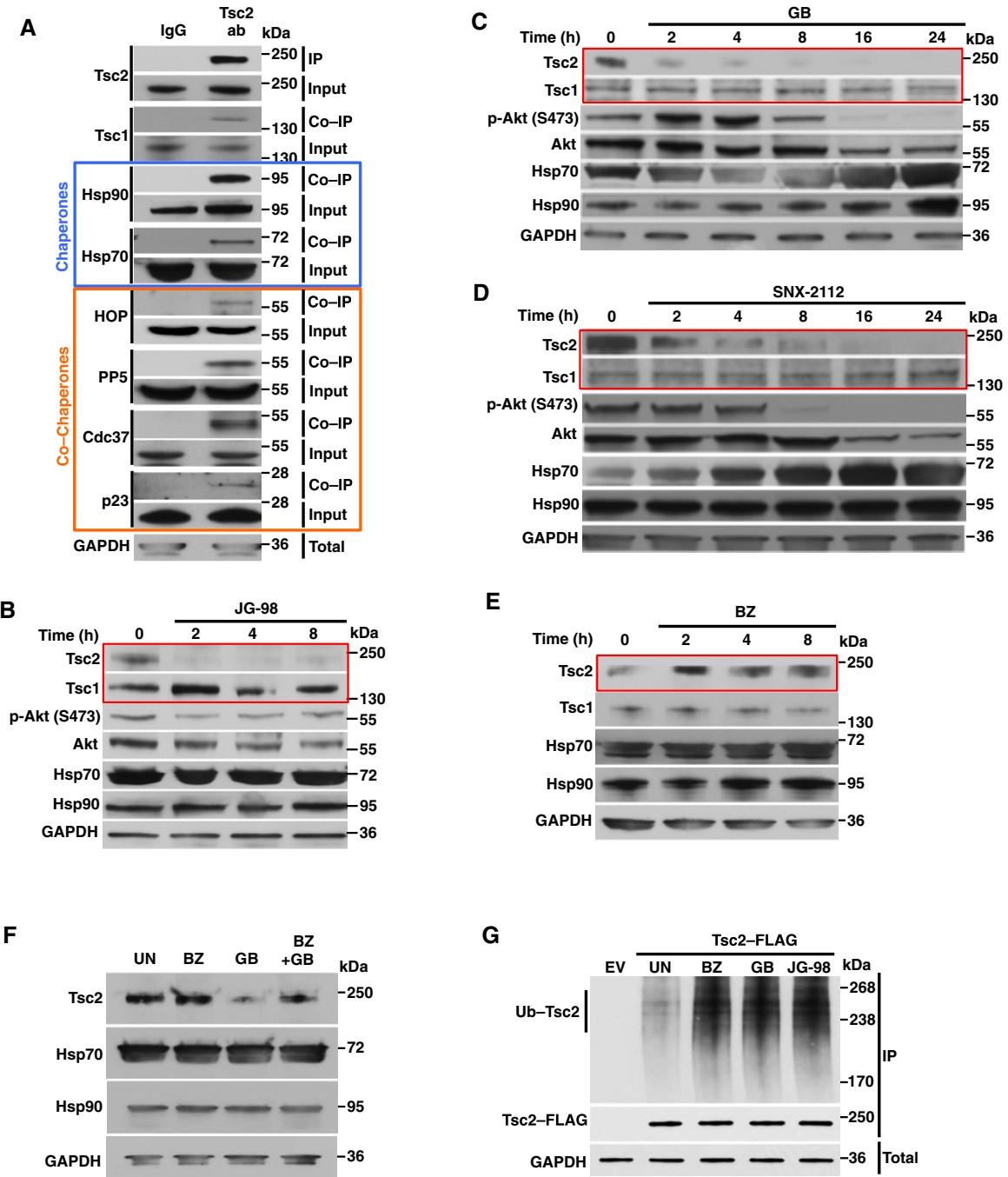

**Figure 1. Tsc2 is a new client of Hsp90.**

A   Tsc2 was immunoprecipitated from HEK293 cell lysates using anti-Tsc2 antibody or IgG (control) and immunoblotted with indicated antibodies to confirm chaperone and co-chaperone interaction.

B   HEK293 cells were treated with 10 μM JG-98 (Hsp70 inhibitor) for the indicated times. Tsc1 protein stability and Tsc2 protein stability were assessed by Western blotting.

C, D   HEK293 cells were treated with either (C) 1 μM GB or (D) 2 μM SNX-2112 (Hsp90 inhibitors) for the indicated times. Tsc1 protein stability and Tsc2 protein stability were assessed by immunoblotting. Akt and phospho-S473-Akt were used as positive controls.

E   HEK293 cells were treated with 50 nM bortezomib (BZ) for the indicated times, and Tsc1 and Tsc2 proteins were evaluated by immunoblotting.

F   50 nM BZ was added to HEK293 cells for 1 h followed by addition of 1 μM GB for 8 h. HEK293 cells were also treated individually with BZ or GB. UN represents untreated cells. Stability of Tsc2 was examined by immunoblotting.

G   Empty vector (EV) or Tsc2-FLAG was used to transiently transfect HEK293 cells for 24 h followed by no treatment (UN), or treatment with either 50 nm BZ, 1 μM GB, or 10 μM JG-98 for 4 h. Tsc2-FLAG was immunoprecipitated, and ubiquitination was examined by immunoblotting with a anti-pan-ubiquitin antibody.

Source data are available online for this figure.

ATPase activity (Fig 2G). Addition of Tsc1-D also significantly inhibited the ATPase activity of Hsp90α (Figs 2G and EV2D). Percentage ATPase activity was based on mmol $P_i$ per mol Hsp90α per minute for Hsp90α alone and Tsc1-D was titrated until inhibition was achieved (Hsp90α = 300 nM, Tsc1-D = 1000 nM). We repeated the experiments in Fig 2G using BSA as a negative control (Fig EV2E).

To obtain further evidence that Tsc1 is a regulator of Hsp90 ATPase activity, we used previously created Cre knockout ($Tsc1^{GFAP}$CKO) mice with conditional inactivation of the *TSC1* gene predominantly in glia (Uhlmann *et al*, 2002; Zeng *et al*, 2008) Brain tissue was harvested from $Tsc1^{GFAP}$CKO ($Tsc1^{flox/flox}$-GFAP-Cre) mice, as well as $Tsc1^{flox/+}$-GFAP-Cre and $Tsc1^{flox/flox}$ littermate controls (Uhlmann *et al*, 2002; Zeng *et al*, 2008). Hsp90 was isolated from $Tsc1^{GFAP}$CKO and control brains (Fig EV2F). PiPer Phosphate Assay was used to measure the isolated Hsp90 ATPase activity. Our data show that the conditional knockout of *TSC1* in mouse brain caused a significant increase in ATPase activity compared to the control samples (Fig 2H). Compromised Hsp90 chaperone function impacts the stability and/or the activity of its client proteins. We therefore examined the stability of a selection of client proteins in lysates prepared from $Tsc1^{GFAP}$CKO and control mouse brains by immunoblotting. Conditional inactivation of Tsc1 in mouse brains caused a down-regulation of the kinase clients such as ErbB2 and Ulk1 and non-kinase client proteins such as estrogen receptor (ER) and GR (Fig 2I). Our data suggest that Tsc1 is a co-chaperone of Hsp90 and a potent inhibitor of the chaperone cycle. Furthermore, Tsc1 co-chaperone can influence and impact the stability and activity of a large number of Hsp90 client proteins.

## Tsc1 co-chaperone enables Tsc2 binding to Hsp90

To gain further insight into Tsc1 function as a new co-chaperone of Hsp90, we examined a selection of its *bona fide* clients in wild-type and *TSC1*-deficient (Tsc1$^{-/-}$) murine embryo fibroblast (MEF) cell lines. Absence of Tsc1 significantly reduced both the activity and the stability of kinase clients phospho-Y416-c-Src, total c-Src, ErbB2, Ulk1, Raf-1, and Cdk4, and non-kinase clients phospho-S211-GR, total GR, FLCN, and Tsc2 (Fig 3A). To determine the effects of Tsc1

over-expression on the Hsp90 clients, HEK293 cells were transiently transfected with either 2 or 4 μg of Tsc1-HA-tagged plasmid. Over-expression of Tsc1 also caused a reduction in the stability of the Hsp90 kinase clients such as ErbB2, Ulk1, and Raf-1 (Fig 3B). Conversely, high levels of Tsc1 increased the total amount of GR (Fig 3B), but did not impact the stability of Cdk4 kinase. It also had a slight positive impact on the stability of both FLCN and Tsc2 (Fig 3B). We next examined the effects of Tsc1 on Hsp90 chaperone function by assessing the steady-state expression of cystic fibrosis transmembrane conductance regulator (CFTR). CFTR is a client protein of Hsp90 (Youker *et al*, 2004; Wang *et al*, 2006) and relies on the Hsp90 chaperone cycle for correct folding (Youker *et al*, 2004; Wang *et al*, 2006). HEK293 cells were transiently co-transfected with CFTR and either pcDNA3 (control) or Tsc1-FLAG plasmids. Immunoblot analysis of these samples using anti-CFTR antibody showed a doublet (Fig 3C). The upper band is the mature Golgi-processed glycoform of CFTR located at the cell surface and the lower band an immature core-glycosylated protein (Fig 3C). Over-expression of Tsc1 significantly increased CFTR protein, suggesting a deceleration in Hsp90 chaperone activity and, consequently, an increase in CFTR expression. Our data provide further evidence that Tsc1 functions as a new co-chaperone of Hsp90, and its levels compromise Hsp90 chaperoning of client proteins.

We next explored the role of Tsc1 in Hsp90-mediated chaperoning of the Tsc2 client protein. We managed to express and purify a small amount of Tsc1 and Tsc2 from HEK293 cells. Tsc1 directly associates with bacterially expressed and purified Hsp90α-His$_6$ (Fig 3D) and Tsc2 (Fig 3E). Surprisingly, direct interaction of Tsc2 and Hsp90α-His$_6$ did not occur *in vitro*; however, pre-incubation of Hsp90α-His$_6$ with Tsc1 facilitated the formation of Hsp90α:Tsc1:Tsc2 complex (Fig 3F). To explore whether Tsc2 may directly interact with Hsp90 in this trimeric complex, we first showed that treating the Tsc1:Hsp90 complex with Hsp90 inhibitor GB did not disrupt Hsp90 interaction with Tsc1 (Fig 3G). The interaction of Tsc1:Tsc2 was also not disrupted with GB (Fig 3H). However, treating the trimeric Tsc1:Tsc2:Hsp90α complex with GB led to dissociation of Tsc2 from the complex (Fig 3I), whereas Tsc1:Hsp90 interaction from these treated samples was unaffected (Fig 3J). It is

---

**Figure 2. Tsc1 co-chaperone inhibits Hsp90 chaperone function.**

A   Tsc1 was immunoprecipitated from HEK293 cell lysates using anti-Tsc1 antibody and immunoblotted with indicated antibodies to confirm chaperone, co-chaperone, and client protein interaction. IgG was used as control.

B   FLAG-tagged Hsp90α N-, M-, and C-domains were transiently expressed and immunoprecipitated from HEK293 cells. Co-immunoprecipitation (Co-IP) of endogenous Tsc1 and Tsc2 was examined by immunoblotting. Empty vector (EV) was used as a control.

C   Hsp90α-FLAG, Hsp90β-FLAG, and their deleted TPR domain-binding site constructs were transiently expressed and immunoprecipitated from HEK293 cells. Interaction of Tsc1 was assessed by immunoblotting. HOP was used as a positive control. Empty vector (EV) was used as a control.

D   FLAG-tagged Tsc1 domains were transiently expressed and isolated from HEK293 cells. Co-IP of endogenous Hsp70 and Hsp90 was assessed by immunoblotting. Empty vector (EV) was used as a control.

E   Tsc1-D-HA was co-expressed with FLAG-tagged Hsp90α N-, M-, or C- domains in HEK293 cells. Different domains of Hsp90α-FLAG were immunoprecipitated, and Tsc1-D-HA interaction was assessed by immunoblotting. Empty vector (EV) was used as a control.

F   Bacterially expressed and purified Tsc1-D-His$_6$ and Hsp90α binding affinity in the presence of AMPPNP, ADP, and GB were measured by fluorescence anisotropy. The error bars represent mean ± SD of three independent experiments.

G   ATPase activity of Hsp90α *in vitro*. Inhibitory effects of purified Tsc1-D-His$_6$ on the ATPase activity of Hsp90α are shown. All the data represent mean ± SD of three biological replicates. A Student's *t*-test was performed to assess statistical significance (**$P < 0.005$, ****$P < 0.0001$).

H   Hsp90 was isolated and ATPase activity was measured from five $Tsc1^{GFAP}$CKO mice with conditional inactivation of the *TSC1* gene predominantly in glia and five $Tsc1^{flox/+}$-GFAP-Cre and $Tsc1^{flox/flox}$ littermate controls (Tsc1-WT). A Student's *t*-test was performed to assess statistical significance (**$P < 0.005$).

I   Tsc1, Hsp90, and client proteins from samples in (H) were examined by immunoblotting. SE (short exposure) and LE (long exposure) of the radiographic film.

Source data are available online for this figure.

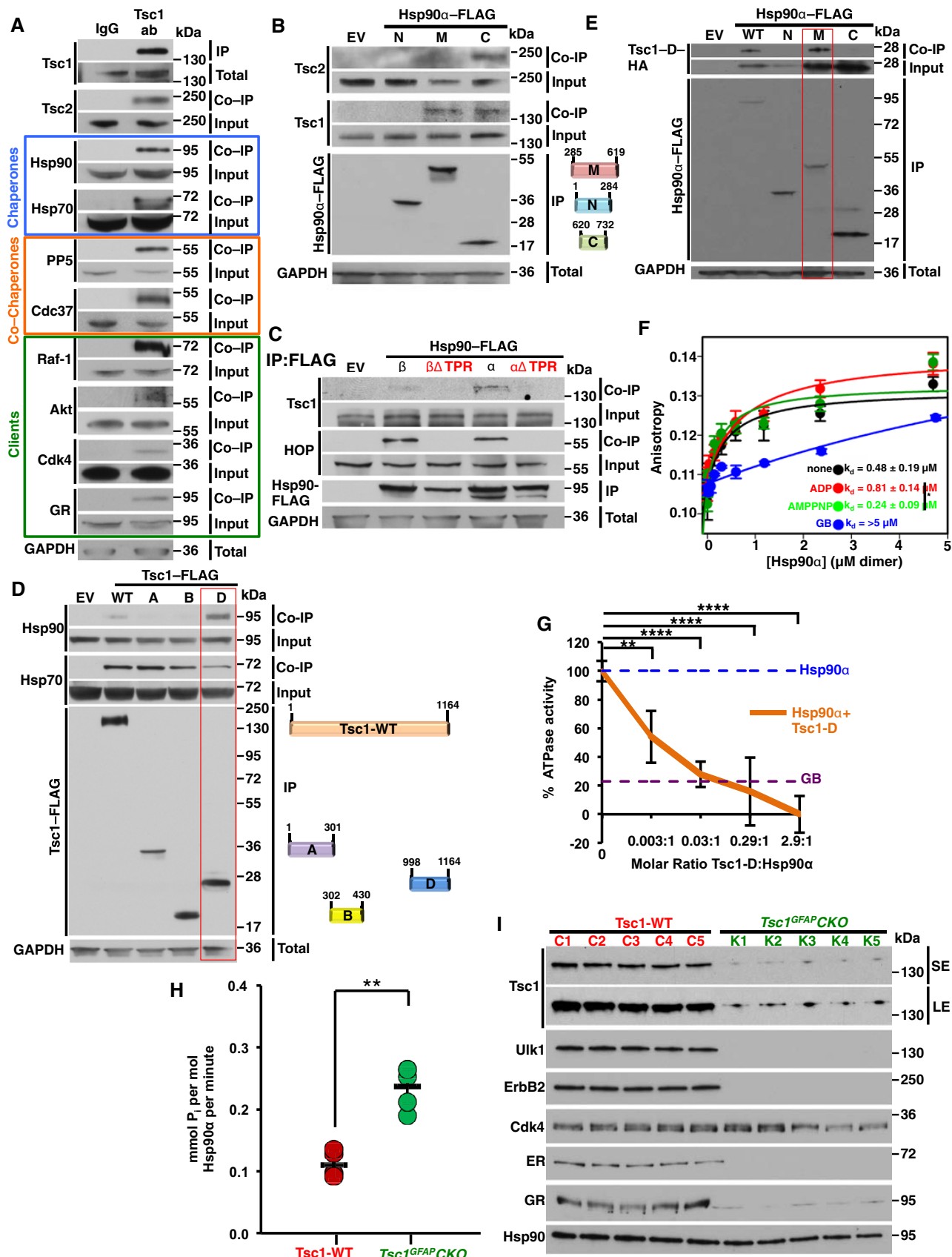

**Figure 2.**

noteworthy that we were unable to produce large amounts of Tsc1 and Tsc2 proteins from bacteria, yeast, or baculovirus expression systems for biophysical analysis. Taken together, our data, at least qualitatively, suggest that the co-chaperone Tsc1 acts as a loading scaffold facilitating the direct binding of Tsc2 to Hsp90 (Fig 3K).

In terms of the clinical relevance of our findings, we explored the pathogenic mutant L117P, which is in the amino-terminus domain of Tsc1 but not within the Tsc2-binding domain. In agreement with previous works (Hoogeveen-Westerveld *et al*, 2010, 2012), the L117P-Tsc1-HA is unstable when transiently expressed in the Tsc1$^{-/-}$ MEF cell line. However, over-expression of this mutant (i.e., using 6 μg L117P-Tsc1-HA plasmid for transfection) slightly stabilized the L117P-Tsc1 as well as Tsc2 in the Tsc1$^{-/-}$ MEF cells (Fig EV3A). We next showed that transfection of the Tsc1$^{-/-}$ MEF cells with either 1 μg of empty vector or L117P-Tsc1-HA plasmid for 24 h and followed by treatment with 50 nM proteasome inhibitor BZ led to increased stability of L117P-Tsc1-HA (Fig EV3B). Finally, we overexpressed (6 μg L117P-Tsc1-HA plasmid) in Tsc1$^{-/-}$ MEF cells followed by immunoprecipitation and detection of Tsc2 and Hsp90 Co-IP by immunoblotting (Fig EV3C). Our findings here suggest that loss of Tsc1 as the result of mutation in its N-domain destabilizes Tsc1 and therefore prevents Tsc2 from binding to Hsp90. This makes Tsc2 susceptible to proteasome-mediated degradation.

### The co-chaperone Tsc1 binds to the closed conformation of Hsp90

Some co-chaperones have been shown to impact the affinity of Hsp90 for binding to ATP or amino-domain inhibitors of Hsp90 both *in vitro* and *in vivo* (Walton-Diaz *et al*, 2013). Here we used the wild-type and Tsc1$^{-/-}$ MEF cell lines and showed that absence of Tsc1 significantly reduces Hsp90 binding to ATP-agarose (Fig 4A). Conversely, over-expression of Tsc1-HA or its Hsp90-binding fragment Tsc1-D-FLAG in HEK293 cells increased Hsp90 binding to ATP-agarose (Fig 4B and C). We obtained similar results with the Hsp90 inhibitor GB, where lack of Tsc1 in MEF cells significantly

reduced the binding of Hsp90 to 1 μM biotinylated GB (Fig 4D). However, over-expression of Tsc1-HA or Tsc1-D-FLAG in HEK293 cells increased the binding of Hsp90 to 0.1 μM biotinylated GB (Fig 4E and F). Taken together, our results suggest that Tsc1 increases the binding of Hsp90 to ATP and its inhibitor GB.

Hsp90 ATPase activity is coupled to a conformational cycle in which the Hsp90 dimer progresses from an N-terminally "open" state to a closed conformation. Human Hsp90α-E47A and D93A mutants can promote either "closed" or "open," respectively (Panaretou *et al*, 1998). These two mutants and wild-type Hsp90α-FLAG were transiently expressed and immunoprecipitated from HEK293 cells. Tsc1 interacts with both Hsp90α-FLAG and the E47A mutant. However, Tsc1 binding with the "open" conformation mutant D93A was reduced (Fig 4G). We repeated these experiments with Tsc1-D-HA and show that this fragment binds with the same affinity to the wild-type Hsp90α-FLAG, E47A, and D93A mutants (Fig 4H). These data suggest that the Tsc1-D fragment, which binds to the Hsp90 middle domain (Fig 2E), associates with Hsp90, and the "open" or "closed" conformations do not impact this interaction.

### Tsc1 dimer binds to both protomers of the Hsp90 middle domain

Our initial data revealed that Tsc1 inhibits Hsp90 chaperone function (Fig 2G) and also that Tsc1 is not in a same complex as Aha1 (Fig EV2A). To gain further insight into the mechanism of Tsc1 competition with Aha1 for binding to Hsp90, we first showed that Hsp90 binds stronger to Aha1 in the absence of Tsc1 in MEF cells (Fig 5A). Our previous work showed that phosphomimetic Hsp90α-Y313E has a stronger affinity for binding to Aha1 than wild-type Hsp90 (Xu *et al*, 2012). Here, we showed that Hsp90α-Y313E does not interact with Tsc1 (Fig 5B). We next explored whether Tsc1 binds to the same amino acids in the middle domain of Hsp90 that are occupied by Aha1. This was achieved using the V410E-FLAG mutant, because this residue is known to participate in the interaction of Aha1-N with the middle domain of Hsp90 (Retzlaff *et al*, 2010). Like Aha1, Tsc1 interaction with Hsp90α-V410E (Fig 5C) was completely abrogated. Previous work has shown that Aha1 binds in

---

**Figure 3. Tsc1 facilitates the chaperoning of Hsp90 clients.**

A   Endogenous protein levels of Hsp90 kinase and non-kinase clients from wild-type and Tsc1$^{-/-}$ MEF cells were assessed by immunoblotting.

B   Transient over-expression of Tsc1-HA in HEK293 cells and its impact on levels of Hsp90 clients was assessed by immunoblotting. Empty vector (EV) was used as a control.

C   HEK293 cells were co-transfected with CFTR and Tsc1-FLAG. CFTR and Tsc1-FLAG were detected by immunoblotting after 24 h. Empty vector (EV) was used as a control.

D   Tsc1 interacts with Hsp90α-His$_6$ *in vitro*. Bacterially expressed and purified Hsp90α-His$_6$ was bound to Ni-NTA agarose and then incubated with 10 ng pure Tsc1. Hsp90α-His$_6$ pulldown and Tsc1 co-pulldown were examined by immunoblotting.

E   Tsc1 binds to Tsc2 *in vitro*. Purified Tsc2 was bound to recombinant protein G agarose using anti-Tsc2 antibody and then incubated with 10 ng pure Tsc1. Tsc2 immunoprecipitation and Tsc1 co-immunoprecipitation were assessed by immunoblotting.

F   Tsc1, Tsc2, and Hsp90α form a trimeric complex *in vitro*. Bacterially expressed and purified Hsp90α-His$_6$ was bound to Ni-NTA agarose and followed by incubation with 10 ng Tsc1 and then 10 ng Tsc2 *in vitro*. Hsp90α-His$_6$ pulldown and Tsc1 and Tsc2 co-pulldown were examined by immunoblotting.

G   Samples from (D) were treated with indicated GB for 1 h at 4°C. Hsp90α-His$_6$ pulldown and Tsc1 co-pulldown were examined by immunoblotting.

H   Samples from (E) were treated with indicated GB for 1 h at 4°C. Tsc2 immunoprecipitation and Tsc1 co-immunoprecipitation were assessed by immunoblotting.

I   Samples from (F) were treated with indicated GB for 1 h at 4°C. Tsc2 immunoprecipitation and Tsc1 and Hsp90α co-immunoprecipitation were assessed by immunoblotting.

J   Supernatant from (I) was used to immunoprecipitate Hsp90α-His$_6$ and co-immunoprecipitate Tsc1, which were examined by immunoblotting.

K   Co-chaperone Tsc1 binds to Tsc2 (1) and acts as a loading scaffold facilitating the direct binding of Tsc2 to Hsp90 (2), eventually leading to dissociation of Tsc1 from Hsp90 (3).

Source data are available online for this figure.

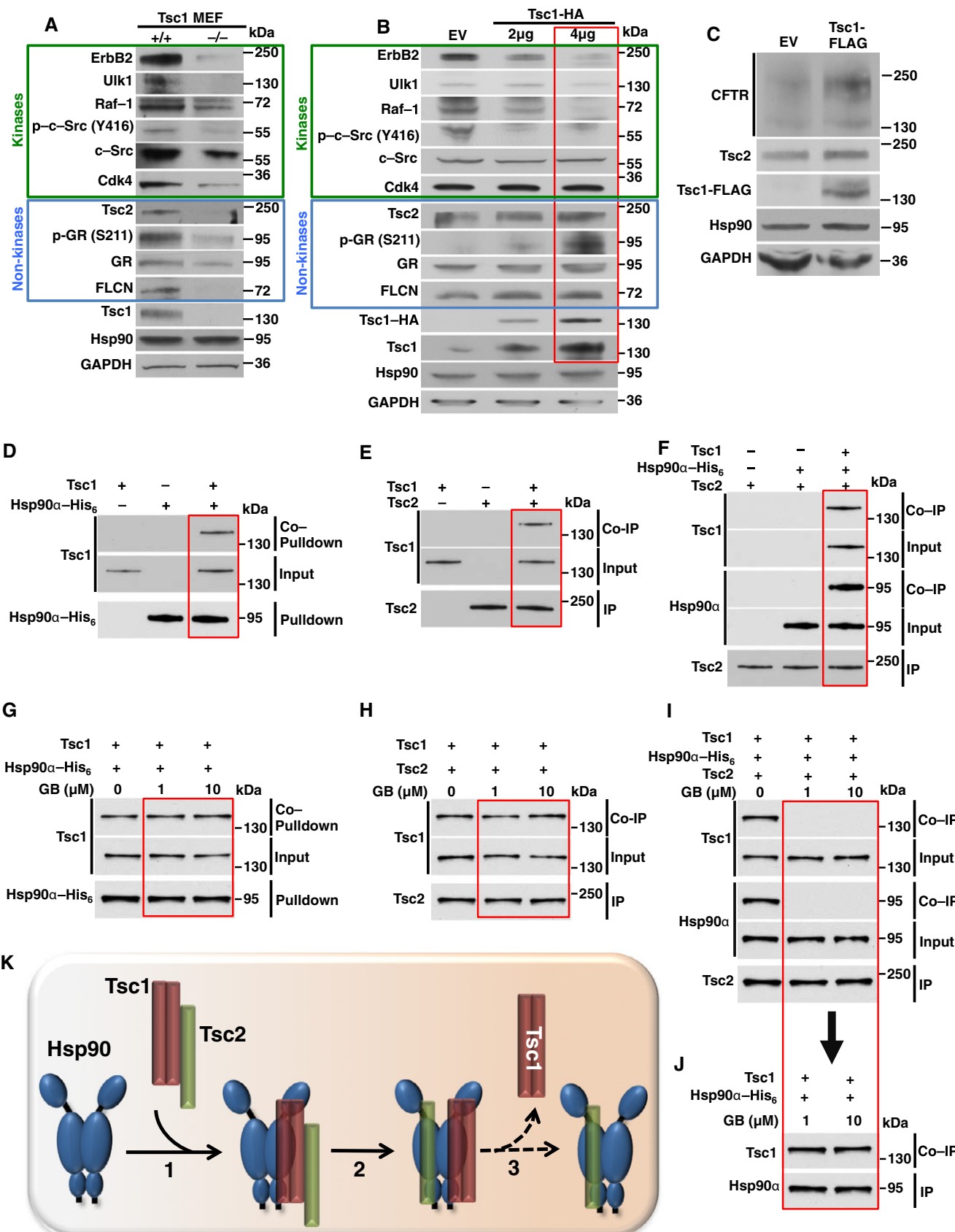

Figure 3.

an asymmetric manner to an Hsp90 dimer (Retzlaff *et al*, 2010). To test whether Tsc1 also binds to Hsp90 in a similar manner as Aha1, we transiently co-expressed both Hsp90α-FLAG and Hsp90α-HA in HEK293 cells. Following immunoprecipitation with anti-FLAG M2 affinity gel, Hsp90α-FLAG was competed from the agarose with FLAG peptide and then subjected to a second round of immunoprecipitation using HA-agarose. The isolated Hsp90 (WT-FLAG:WT-HA) is a homodimer with one protomer tagged with FLAG and the other protomer with HA. We were able to co-immunoprecipitate

Tsc1 and Aha1 with Hsp90α-FLAG:HA (Fig 5D). We also carried out similar experiment with the exception of co-expressing Hsp90α-V410E-FLAG with Hsp90α-HA in HEK293 cells. The isolated Hsp90α (V410E-FLAG:WT-HA) consists of an Hsp90α-FLAG-V410E protomer and a second subunit with the wild-type Hsp90α-HA (Fig 5D). We co-immunoprecipitated Aha1 in this sample, which is in agreement with Aha1 interaction with one subunit of the Hsp90 dimer (Fig 5D; Retzlaff *et al*, 2010; Mollapour *et al*, 2014). Tsc1 however did not bind to Hsp90α-V410E-FLAG:WT-HA, suggesting

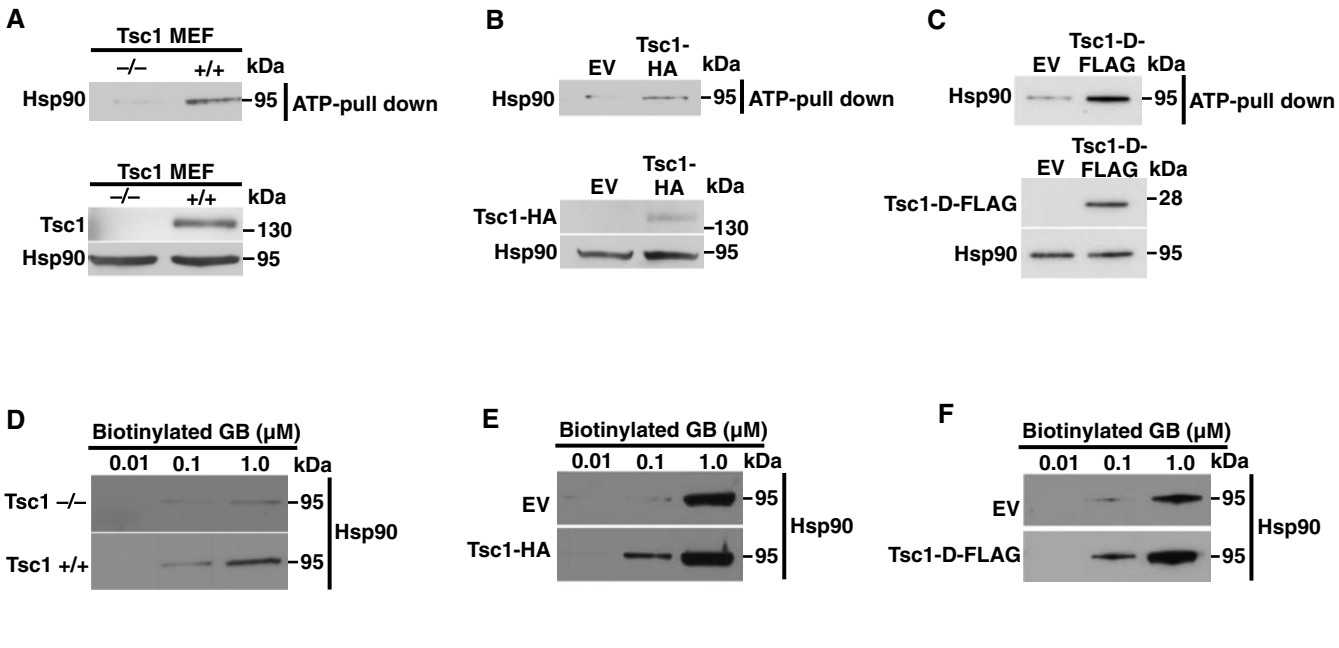

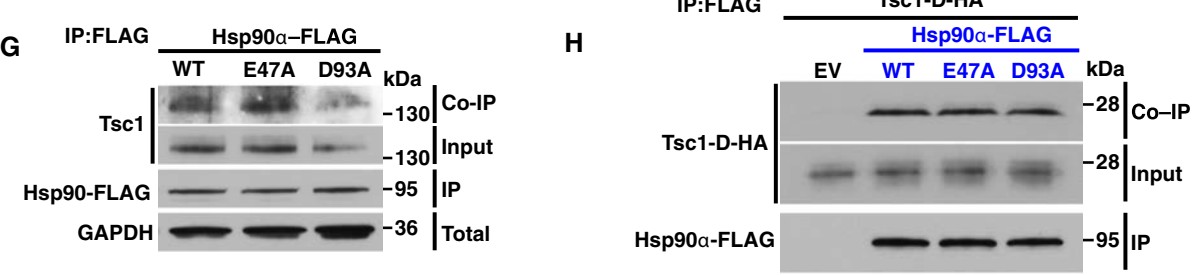

**Figure 4.  Tsc1 increases Hsp90 binding to ATP and drugs.**

A   Lysates from wild-type and Tsc1$^{-/-}$ MEF cells were incubated with ATP-agarose. Hsp90 binding to ATP-agarose was examined by immunoblotting.

B   Tsc1-HA or empty vector (EV) was transiently expressed in HEK293 cells for 24 h followed by incubating the lysates with ATP-agarose. Hsp90 binding to ATP-agarose was examined by immunoblotting.

C   Tsc1-D-FLAG or empty vector (EV) was transiently expressed in HEK293 cells for 24 h. Lysates were incubated with ATP-agarose. Hsp90 binding to ATP-agarose was examined by immunoblotting.

D   Lysates from (A) were incubated with indicated amounts of biotinylated GB followed by streptavidin agarose beads. Hsp90 was detected by immunoblotting.

E   Lysates from (B) were incubated with indicated amounts of biotinylated GB followed by streptavidin agarose beads. Hsp90 was detected by immunoblotting.

F   Lysates from (C) were incubated with indicated amounts of biotinylated GB followed by streptavidin agarose beads. Hsp90 was detected by immunoblotting.

G   Hsp90α-FLAG WT and its "open" D93A or "closed" E47A mutants were transiently expressed in HEK293 cells. Hsp90-FLAG proteins were immunoprecipitated, and co-immunoprecipitation of the endogenous Tsc1 was examined by immunoblotting.

H   Hsp90α-FLAG, D93A, and E47A mutants were co-expressed with Tsc1-D-HA in HEK293 cells. Hsp90-FLAG and its mutants were immunoprecipitated, and co-immunoprecipitation of Tsc1-D-HA was examined by immunoblotting.

Source data are available online for this figure.

that Tsc1 interacts symmetrically with both subunits of the Hsp90 dimer.

Our initial data identified that region D within Tsc1 (Tsc1-D) binds to the middle domain of Hsp90α (Fig 2E). We next confirmed that as with the full-length Tsc1, Tsc1-D does not co-immunoprecipitate Aha1 from HEK293 cells (Fig 5E). We also showed that Tsc1-D exists as a homodimer, by first co-expressing Tsc1-D-FLAG and Tsc1-D-HA in HEK293 cells. Immunoprecipitation of Tsc1-D-FLAG led to co-immunoprecipitation of Tsc1-D-HA (Fig 5F). We next bacterially expressed and purified Tsc1-D-His$_6$. Gel filtration of the purified protein indicates that Tsc1-D-His$_6$ exists as a dimer in solution, as it migrates with an apparent size of approximately twice the predicted molecular weight of the monomer (22.1 kDa; Fig 5G).

To dissect the competitive binding of Aha1 and Tsc1 to Hsp90, we first bound bacterially expressed and purified Tsc1-D-His$_6$ to Ni-NTA agarose and then incubated with 100 ng recombinant Hsp90α. After washing the Ni-NTA agarose with the wash buffer, we added different amounts of recombinant Aha1-FLAG. We found that addition of 200 ng recombinant Aha1-FLAG completely disrupted Hsp90α and Tsc1-D-His$_6$ interaction (Fig 5H). We next used a similar experiment by first binding Aha1-FLAG to anti-FLAG M2 affinity gel followed by addition of 100 ng recombinant Hsp90α to form an Aha1:Hsp90α complex. After washing the agarose beads with the wash buffer, different amounts of Tsc1-D-His$_6$ were added to the complex. Our data show that addition of 1 ng Tsc1-D-His$_6$ was sufficient to completely disrupt the Aha1:Hsp90α complex (Fig 5I). These data indicate that Tsc1 has a stronger affinity for association with Hsp90 than Aha1. Lastly, we tested the simultaneous impact of both Tsc1 and Aha1 on Hsp90 ATPase activity. Our data show that Aha1 stimulates and Tsc1 inhibits Hsp90α ATPase (Figs 5J and EV4A and B). We also show that addition of 1.3 μM Aha1 to Hsp90α after incubation with 1 μM Tsc1 was capable of stimulating Hsp90α ATPase to approximately 100% (Figs 5J and EV4B). Our data suggest that Tsc1 and Aha1 compete for binding to the same sites in the middle domain of Hsp90 in order to fine-tune its ATPase activity.

## Phosphorylation of Aha1 displaces Tsc1 from Hsp90

We previously showed that c-Abl-mediated phosphorylation of Aha1-Y223 promotes its interaction with Hsp90 (Dunn *et al*, 2015). We therefore examined the influence of phospho-Aha1-Y223 on Tsc1 binding to Hsp90. We bound Tsc1-D-His$_6$ to Ni-NTA agarose and incubated with 100 ng recombinant Hsp90α. After washing the agarose with buffer, we added different amounts of Aha1-Y223E-FLAG. Our data show that addition of 1 ng of phospho-mimetic Y223E mutant completely dissociated Hsp90α from Tsc1-D-His$_6$ (Fig 6A) as opposed to 200 ng WT Aha1 (Fig 5H). Tsc1 has no similarities with other proteins and although orthologs can be found in most eukaryotic cells, including the fission yeast *Schizosaccharomyces pombe*, no *TSC1* gene exists in *Saccharomyces cerevisiae*. We used this system to express *TSC1-FLAG* under galactose-inducible promoter *GAL1* in yeast containing either human Hsp90α or yeast Hsp82 (yHsp90) as the sole functional Hsp90. Tsc1-FLAG was detectable only in *S. cerevisiae* containing Hsp90α (Fig 6B) and this expression caused lethality in yeast cells (Fig 6C). We next used this strain to transform an empty vector (ADH-pRS424) or human Aha1 or its phosphomimetic Y223E under alcohol dehydrogenase 1 (*ADH1*) promoter. Consistent with our previous work (Dunn *et al*, 2015), immunoprecipitation of Hsp90α led to co-immunoprecipitation of human Aha1 and strongly to Aha1-Y223E (Fig 6D). However, growing these yeast cells on galactose media and consequent expression of Tsc1-FLAG led to its association with Hsp90α in the absence of human Aha1 (Fig 6D). Presence of human Aha1 slightly reduced the interaction of Tsc1-FLAG with Hsp90α and this association was completely abrogated in the presence of the phosphomimetic Aha1-Y223E mutant (Fig 6D). This mutant also rescued the toxic effect of Tsc1-FLAG in yeast cells (Fig 6E).

We obtained further evidence by treating the Tsc1$^{-/-}$ and wild-type MEF cell lines with 20 μM 5-(1,3-di-aryl-1H-pyrazol-4-yl) hydantoin, 5-[3-(4-fluorophenyl)-1-phenyl-1H-pyrazol-4-yl]-2,4-imidazolidinedione (DPH), which binds to the myristoyl binding site of c-Abl and leads to its activation (Yang *et al*, 2011). As expected,

**Figure 5. Tsc1 dimer binds to both protomers of Hsp90 middle domain.**

A　Hsp90 was isolated from wild-type and Tsc1$^{-/-}$ MEF cells lysates using anti-Hsp90α/β antibody and immunoblotted with Aha1 antibody.

B　Hsp90α-FLAG and the phosphomutants Y313F and Y313E were transiently expressed and isolated from HEK293 cells. Co-immunoprecipitation of Tsc1 and Aha1 was examined by immunoblotting.

C　Hsp90α-FLAG and V410E mutant were transiently expressed and isolated from HEK293 cells. Co-immunoprecipitation of Tsc1 and Aha1 was examined by immunoblotting.

D　Hsp90α-FLAG and Hsp90α-HA were co-expressed in HEK293 cells. Hsp90α-FLAG was first isolated by anti-FLAG M2 affinity gel and then competed with FLAG peptide. Hsp90α-FLAG:Hsp90α-HA heterodimer protein was isolated by second immunoprecipitation using HA-agarose. Co-immunoprecipitation of Tsc1 and Aha1 was examined by immunoblotting.

E　HEK293 cells transiently expressed Tsc1-D-FLAG. Immunoprecipitation of Tsc1-D-FLAG and co-immunoprecipitation of Hsp90 and Aha1 were examined by immunoblotting.

F　Tsc1-D-FLAG and Tsc1-D-HA were co-expressed in HEK293 cells. Tsc1-D-FLAG was immunoprecipitated, and co-immunoprecipitation of Tsc1-D-HA was examined by immunoblotting.

G　Bacterially expressed Tsc1-D-His$_6$ was concentrated to 10 mg/ml and subjected to gel filtration using a Superdex 75 column. The protein migrates as a monodisperse peak with an apparent MW of ~52 kDa.

H　Tsc1 and Aha1 compete for binding to Hsp90α. Tsc1-D-His$_6$ was attached to Ni-NTA agarose and then incubated with Hsp90α. Ni-NTA agarose was then washed and incubated with the indicated amounts of Aha1-FLAG.

I　Aha1-FLAG attached to anti-FLAG M2 affinity gel was incubated with Hsp90α initially and then washed and incubated with indicated amounts of the Tsc1-D-His$_6$.

J　Tsc1-D-His$_6$ inhibits Hsp90α ATPase activity after 30 min. Addition of 1.3 μM Aha1-FLAG stimulated the ATPase activity. All the data represent mean ± SD of three biological replicates. A Student's *t*-test was performed to assess statistical significance (*P < 0.05, ***P < 0.0005).

Source data are available online for this figure.

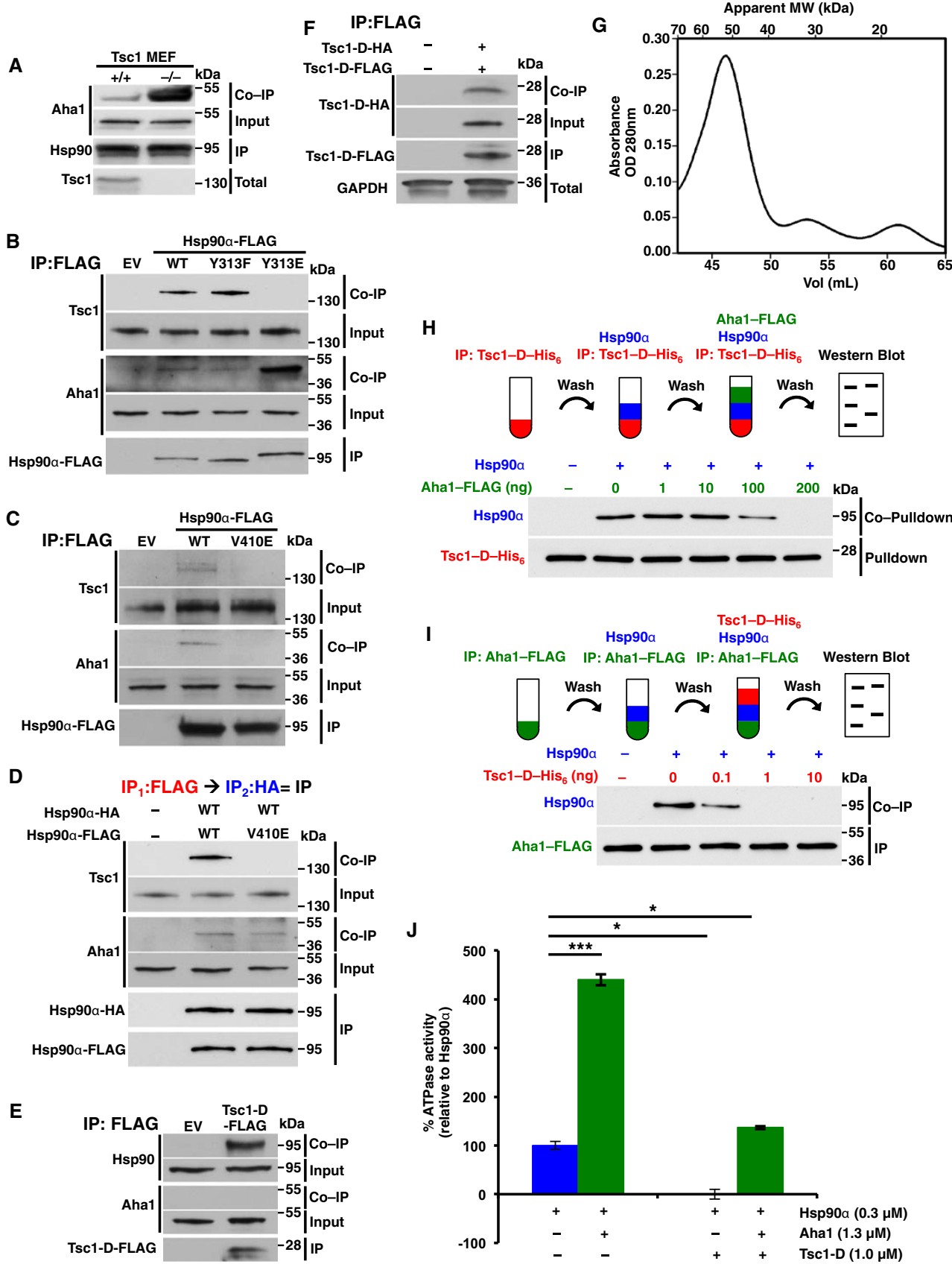

**Figure 5.**

Hsp90 binds to both Aha1 and Tsc1 (presumably separate complexes) in the wild-type MEF cells (Fig 6F). DPH-mediated activation of c-Abl and hyperphosphorylation of Aha1 in the wild-type MEF cells increased Hsp90 binding to Aha1 but abrogated its interaction with Tsc1 (Fig 6F). Hsp90 also bound stronger to Aha1 in Tsc1$^{-/-}$ MEF cells (Fig 6F). We repeated the same experiment but treated the MEF cells with 5 μM GNF-5 (a selective allosteric c-Abl inhibitor) for 24 h (Yang *et al*, 2011). Inhibition of c-Abl in wild-type MEF cells reduced phosphorylation of Aha1 and its association with Hsp90 and led to enhanced interaction of Tsc1 with Hsp90 (Fig 6G). Interestingly, Aha1 did not dissociate from Hsp90 in Tsc1$^{-/-}$ MEF cells treated with GNF-5 (Fig 6G). These data suggest that Tsc1 plays an essential role in dissociation of Aha1 from Hsp90 in cells.

We further explored the role of c-Abl-mediated phosphorylation of Aha1 and its competition with Tsc1 for binding to Hsp90 by using the c-Abl$^{-/-}$ MEF cells. Absence of c-Abl in this MEF cell line, and therefore lack of Aha1-Y223 phosphorylation, abrogated Aha1 interaction with Hsp90, which is consistent with our previous work (Dunn *et al*, 2015; Fig 6H). Conversely, Tsc1 binding to Hsp90 was enhanced in the c-Abl$^{-/-}$ MEF cells compared to the wild-type MEF cells (Fig 6H). Lastly, silencing of *TSC1* by siRNA in c-Abl$^{-/-}$ MEF cells restored Aha1 binding with Hsp90 (Fig 6I, Appendix Table S1). Taken together, our data here suggest a dynamic equilibrium in binding of Aha1 and Tsc1 to Hsp90 that can be influenced in part by phosphorylation of Aha1.

## Discussion

Mutations in either the *TSC1* or *TSC2* tumor suppressor genes cause TSC, which is a genetic syndrome that leads to manifestation of benign tumors in the brain and other organs such as kidneys, heart, eyes, lungs, and skin. Tsc1 and Tsc2 form a physical and functional complex that ultimately inhibits mTOR. Tsc2 is a GTPase-activating protein, and Tsc1 prevents Tsc2 interaction with HERC1 ubiquitin ligase (Crino *et al*, 2006; Henske *et al*, 2016). Previous work has also shown that Tsc1 and Tsc2 both interact with the co-chaperone complex R2TP/Prefoldin-like (R2TP/PFDL) complex (Cloutier *et al*,

2017). In this study, we demonstrate that Tsc2 is a client of the molecular chaperones Hsp70 and Hsp90, since treating the cells with Hsp70 or Hsp90 inhibitors leads to Tsc2 ubiquitination and proteasomal degradation. We further demonstrate that Tsc1 is a new co-chaperone of Hsp90. Tsc1 carboxy-domain (998–1,164 aa; Tsc1-D) exists as a homodimer and interacts with the middle domain of Hsp90. 100 nM of Tsc1-D is sufficient to inhibit or decelerate Hsp90 ATPase activity by 72%. Conversely, Hsp90 isolated from mouse brains with conditional inactivation of Tsc1 had a higher ATPase activity compared to the Hsp90 from control samples. Furthermore, inactivation of Tsc1 in these samples caused a down-regulation of the kinase clients such as ErbB2 and Ulk1 as well as non-kinase client proteins such as estrogen receptor (ER) and GR. This is consistent with our findings obtained from the Tsc1$^{-/-}$ MEF cells, which showed down-regulation of kinase clients such as ErbB2, Ulk1, c-Src, Raf1, and Cdk4 and non-kinase clients such as GR, FLCN, and Tsc2. Over-expression of Tsc1 in HEK293 cells also down-regulated the kinase clients but surprisingly stabilized the steroid hormone receptors and non-kinase clients such as FLCN and CFTR possibly because of the slow chaperone cycle of Hsp90. Our findings suggest that the newly identified function of Tsc1 is to regulate the chaperoning of Hsp90 clients including Tsc2. Like other co-chaperones such as Cdc37 and HOP, Tsc1 behaves as a scaffold in order to load clients, in this case Tsc2, to Hsp90. It is also conceivable that Tsc1 regulates the chaperone function of Hsp70 and perhaps like HOP, connects the transfer of clients between Hsp70 and Hsp90.

Previous works have shown that pathogenic missense mutations of Tsc1 in its N-domain can destabilize the Tsc1 protein (Hoogeveen-Westerveld *et al*, 2010, 2012). We have shown that the pathogenic mutant L117P-Tsc1 is unstable and also prevents Tsc2 from binding to Hsp90. This leads to destabilization of Tsc2 and perhaps proteasome-mediated degradation.

What is the mechanism of Tsc1 interaction with and inhibition of Hsp90? Hsp90 chaperone cycle is linked to its ability to bind and hydrolyze ATP (Panaretou *et al*, 1998). This provides a series of conformational changes in which the Hsp90 dimer progresses from an N-terminally "open" state to a "closed" conformation. Tsc1 prefers binding to the closed conformation mutant E47A rather than the open state mutant D93A. While the Tsc1-D fragment interacts

---

**Figure 6. Phosphorylation of Aha1 displaces Tsc1 from Hsp90.**

A   Tsc1-D-His$_6$ was attached to Ni-NTA agarose and then incubated with Hsp90α. Ni-NTA agarose was then washed and incubated with the indicated amounts of phosphomimetic Aha1-Y223E-FLAG.

B   PP30 yeast strain expressing either yHsp90 or Hsp90α as the sole copies of Hsp90 was transformed by *GAL1-TSC1-FLAG* and grown on raffinose (Raf) overnight. The cells were shifted to galactose (Gal) media for 4 h. Expression of Tsc1-FLAG was assessed by immunoblotting. Sba1 was used as a loading control.

C   Strains in (B) were spotted at 1:10 dilution series on YPED or YPGal media. Plates were incubated at 30°C for 2 days.

D   Yeast strains in (B) were also transformed with *ADH1-AHA1* or the phosphomimetic mutant Y223E. Cells were grown in similar conditions as (B) and the expression of Aha1, Tsc1-FLAG, and Hsp90α was examined by immunoblotting.

E   Yeast strains from (D) were spotted at 1:10 dilution series on YPED or YPGal media. Plates were incubated at 30°C for 2 days.

F   Wild-type and Tsc1$^{-/-}$ MEF cells were treated with 20 μM DPH (c-Abl activator) for 6 h prior to lysis. Hsp90 (Hsp90α/β) was immunoprecipitated, and co-immunoprecipitation of Aha1 and Tsc1 was examined by immunoblotting.

G   Wild-type and Tsc1$^{-/-}$ MEF cells were treated with 5 μM GNF-5 (c-Abl inhibitor) for 24 h prior to lysis. Hsp90 (Hsp90α/β) was immunoprecipitated, and co-immunoprecipitation of Aha1 and Tsc1 was examined by immunoblotting.

H   Hsp90 (Hsp90α/β) was immunoprecipitated from wild-type and c-Abl$^{-/-}$ MEF cell lysate. Co-immunoprecipitation of Aha1 and Tsc1 was examined by immunoblotting.

I   Hsp90 (Hsp90α/β) was immunoprecipitated from c-Abl$^{-/-}$ MEF with siRNA knockdown of *TSC1*. Co-immunoprecipitation of Aha1 and Tsc1 was examined by immunoblotting.

Source data are available online for this figure.

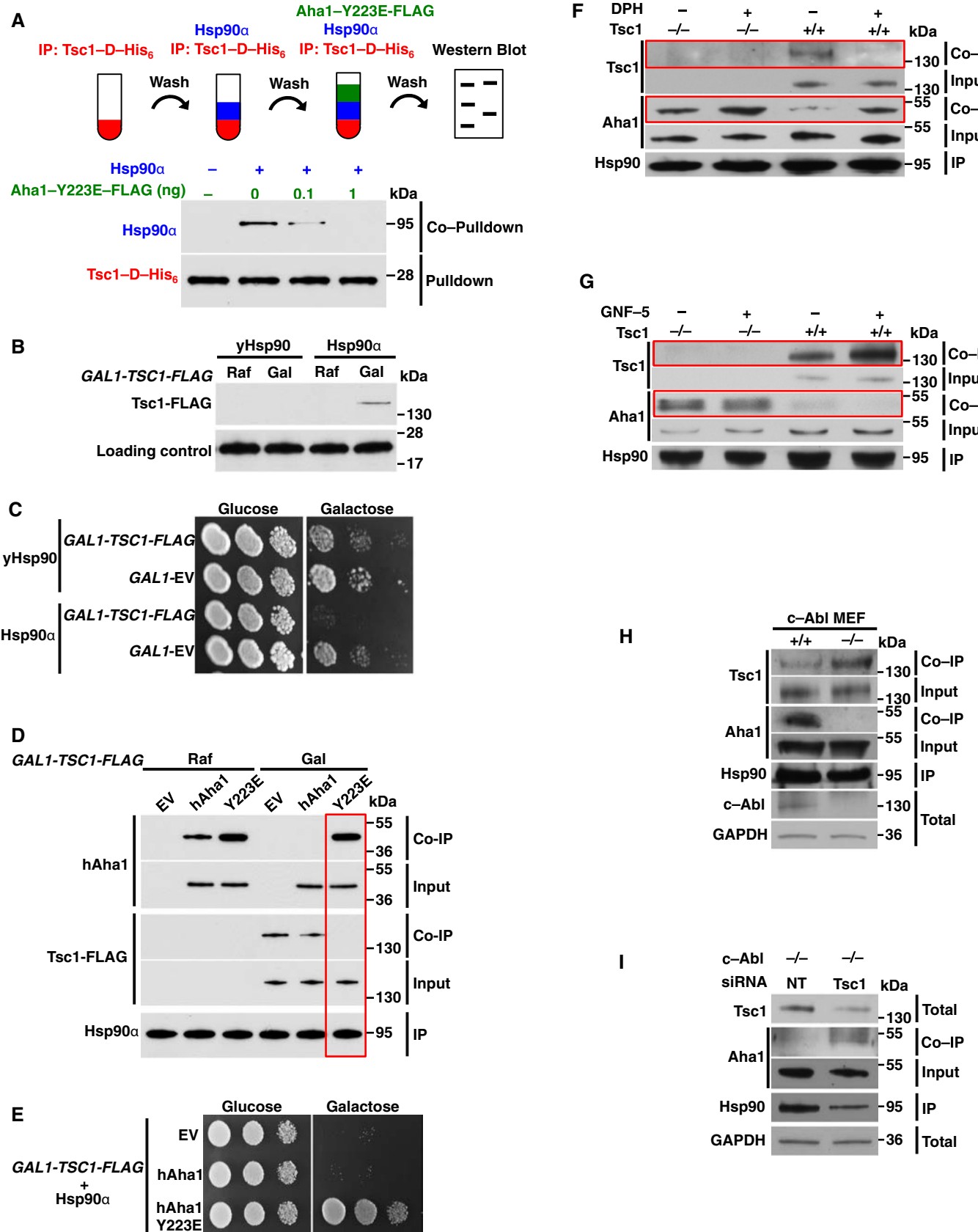

**Figure 6.**

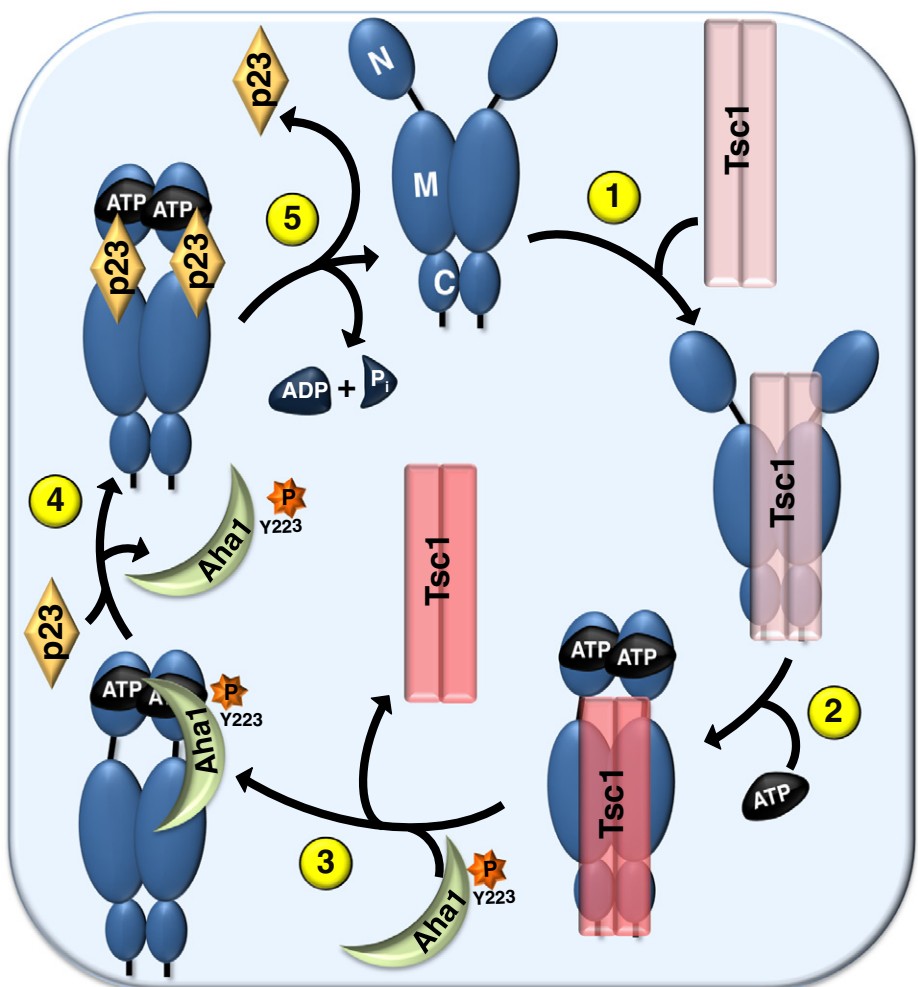

**Figure 7. Impact of Tsc1 co-chaperone on the Hsp90 chaperone cycle.**

(1) Tsc1 binds weakly to the Hsp90 "open" conformation in the absence of nucleotides. (2) Upon ATP binding to the N-domain of Hsp90, Tsc1 interaction strengthens to the "closed" conformation of Hsp90. (3) c-Abl-mediated phosphorylation of Aha1-Y223 displaces Tsc1 from Hsp90. (4) Later in the chaperone cycle, p23 is predicted to expel phosphorylated Aha1-Y223 from Hsp90, (5) allowing Hsp90 to hydrolyze ATP.

with the middle domain of Hsp90, deletion of the MEEVD motif (the TPR domain-binding site) at the very C-terminus of Hsp90 completely abrogates Tsc1 binding to Hsp90. Our data suggest that the binding of Tsc1 to Hsp90 is complex, and it requires multiple surfaces of Tsc1 to interact with Hsp90. Tsc1 is also not in the same Hsp90 complex with Aha1, p23, or HOP co-chaperones. Because Tsc1 and the co-chaperone Aha1 both bind the middle domain and do not exist in the same Hsp90 complex, we examined Aha1 binding site mutants for Tsc1 interaction. Hsp90α-V410 participates in the interaction of Aha1-N with the middle domain of Hsp90α (Meyer *et al*, 2004). The Hsp90α-V410E mutant does not bind to either Aha1 or Tsc1. Interestingly, although V410E mutation of only one protomer of the Hsp90 middle domain binds to Aha1, this mutant does not interact with Tsc1. We conclude that a Tsc1 dimer needs to bind to both protomers of the Hsp90 middle domain. This observation also provides an explanation for Tsc1 competing with and masking Aha1 and p23 binding to Hsp90 (Fig 7). Because Tsc1

interacts with the Aha1 binding sites on Hsp90 middle domain, we predict that Tsc1 prevents the conformational changes in the catalytic loop of Hsp90 that stops the release of R400 from its retracted inactivating conformation consequently inhibiting Hsp90 ATPase activity.

Tsc1 has a high affinity for Hsp90 and significantly increases Hsp90 binding to nucleotides or N-domain inhibitors. In contrast, almost 200-fold more Aha1 protein is required to displace Tsc1-D from Hsp90 *in vitro*. We have previously shown that c-Abl phosphorylation of Aha1-Y223 increases Aha1 association with Hsp90 (Dunn *et al*, 2015). The phosphomimetic Aha1-Y223E binds with similar affinity as Tsc1 to Hsp90, therefore providing a mechanism for equilibrium between these two co-chaperones with respect to binding to Hsp90 (Fig 7). These results establish an active role for Tsc1 as a facilitator of Hsp90 chaperoning of kinase and non-kinase clients, including Tsc2 therefore preventing their ubiquitination and degradation in the proteasome.

This work provides an explanation of how Tsc1 is involved in the stability of Tsc2 specifically as well as a broader function for Tsc1 as a new co-chaperone of Hsp90. We show here that loss of Tsc1 leads to dysregulation of Hsp90 chaperone function and instability of its client proteins, including Tsc2. It is unclear what effect inactivation of Tsc1 and the resultant hyperactive Hsp90 have on maintaining protein homeostasis in tumors in TSC. This crucial point in TSC biology warrants further study.

# Materials and Methods

### Mice

All the mice experiments were performed under the ethical guidelines of the Washington University School of Medicine, and animal protocols were reviewed and approved by the Washington University School of Medicine Institutional Animal Care and Use Committee (IACUC #A-3381-01; Protocol #20160091). All mice used in this study were obtained from an existing breeding colony of $Tsc1^{GFAP}CKO$ mice (Uhlmann *et al*, 2002; Zeng *et al*, 2008) in the animal facility of the Washington University School of Medicine.

### Plasmids

pcDNA3–Tsc1–HA and pcDNA3–Tsc2–FLAG vectors were purchased from Addgene. Tsc1-FLAG was subsequently subcloned into pcDNA3.1 and the 2 μ yeast expression plasmid pYES2 (Appendix Table S1). Primers for the creation of Tsc1 domain-specific plasmids and L117P point mutation can also be found in the primer table. The empty vectors (EV) containing these tags were used and also referred to as controls. Hsp90 mutants lacking the TPR domain and point mutants were developed by PCR (Appendix Table S1). Hsp90α domain-specific plasmids were previously generated by M.R.W. (Woodford *et al*, 2016b). Human Aha1 and Aha1-Y223E mutant were previously cloned into pcDNA3 (Dunn *et al*, 2015). They were subcloned into 2 μ yeast expression plasmid p424ADH (ATCC® 87373™).

### Yeast strains and growth media

Yeast strains listed in Appendix Table S1 were grown on YPDA (2% (w/v) Bacto peptone, 1% yeast extract, 2% glucose, 20 mg/l adenine), YPGal (2% (w/v) Bacto peptone, 1% yeast extract, 2% galactose, 20 mg/l adenine), and YPRaf (2% (w/v) Bacto peptone, 1% yeast extract, 2% raffinose, 20 mg/l adenine). Selective growth was on dropout 2% glucose (DO) medium with appropriate amino acids (Adams *et al*, 1997). Medium pH was adjusted to 6.8 with NaOH before autoclaving.

### Mammalian cell culture

Cultured cell lines human embryonic kidney (HEK293), wild-type, $Tsc1^{-/-}$ cells, and c-Abl$^{-/-}$ MEF cells were grown in Dulbecco's modified Eagle's medium (DMEM, Sigma-Aldrich) supplemented with 10% fetal bovine serum (FBS, Sigma-Aldrich). HEK293 were acquired from American Type Culture Collection (ATCC). All cell lines were maintained in a CellQ incubator (Panasonic Healthcare) at 37°C in an atmosphere containing 5% $CO_2$.

### Bacterial expression and protein purification

Hsp90α-His$_6$ was expressed and purified as previously described (Woodford *et al*, 2016a). The plasmid coding for Tsc1-D was transformed into *E. coli* strain Tuner pLysS and grown on agar plates containing Luria broth (LB), 34 mg/l chloramphenicol, and 50 mg/l ampicillin overnight. Colonies were picked and grown in LB with 34 mg/l chloromphenicol and 200 mg/l ampicillin at 37°C with continuous 200 rpm shaking until $OD_{600} = 0.6$. The temperature was then dropped to 18°C and cells were induced with 100 mg/l IPTG for 18 h. Cells were then harvested by centrifugation, lysed enzymatically, and purified by Ni$^{2+}$-NTA chromatography (Qiagen) using a single step gradient (20 mM Tris pH 7.5, 300 mM NaCl, 10 mM β-ME, 10–250 mM imidazole). The protein was then dialyzed against 20 mM Tris pH 7.5, 10 mM 10 mM β-ME at 4°C overnight. A mild precipitate formed, which was removed by centrifugation. Proteins were > 90% pure by SDS–PAGE. Concentrations were determined using calculated extinction coefficients as previously described (Gasteiger *et al*, 2003). Proteins were flash-frozen on dry ice and stored at −80°C until use.

### Protein extraction, IP, and immunoblotting

Protein extraction from both yeast and mammalian cells was carried out using methods previously described (Woodford *et al*, 2016b). For immunoprecipitation, mammalian cell lysates were incubated with anti-FLAG antibody-conjugated beads (Sigma) for 2 h at 4°C. Pulldowns were achieved by incubating lysate with Ni-NTA agarose (Qiagen) for 2 h at 4°C, or with anti-Hsp90 antibody (835-16F1, Enzo Life Sciences), or Tsc1 or Tsc2 (Cell Signaling) for 1 h followed by protein G agarose for 2 h at 4°C. Immunopellets were washed four times with fresh lysis buffer (20 mM HEPES (pH 7.0), 100 mM NaCl, 1 mM MgCl$_2$, 0.1% NP-40, protease inhibitor cocktail (Roche), and PhosSTOP (Roche)) and eluted in 5× Laemmli buffer. For purification, immunopellets were washed four times with fresh high salt lysis buffer (20 mM HEPES (pH 7.0), 500 mM NaCl, 1 mM MgCl$_2$, 0.1% NP-40, protease inhibitor cocktail (Roche), and PhosSTOP (Roche)) and subsequently competed off with 3× FLAG peptide (Sigma-Aldrich). Proteins bound to Ni-NTA agarose were washed with 50 mM imidazole in lysis buffer [20 mM Tris–HCl (pH 7.5), 100 mM NaCl, protease inhibitor cocktail (Roche), and PhosSTOP (Roche)] and eluted with either 300 mM imidazole in lysis buffer or with 5× Laemmli buffer. Precipitated proteins were separated by SDS–PAGE and transferred to nitrocellulose membranes as previously described (Woodford *et al*, 2016b). Co-immunoprecipitated proteins were detected by immunoblotting as previously demonstrated (Woodford *et al*, 2016b). Dilutions of antibodies are indicated in Appendix Table S1.

### Protein labeling and $K_d$ measurements

Protein labeling and $K_d$ measurements were carried out using methods previously described (Woodford *et al*, 2016b). Tsc1-D was desalted into 20 mM Tris pH 7.5, 150 mM NaCl, and 1 mM TCEP using a PD-10 desalting column (GE Healthcare Bio-Sciences Marlborough, MA) and labeled with Texas Red C2 maleimide per the manufacturer's instructions (Thermo Fisher Scientific, Grand Island,

NY). Labeling stoichiometry was 0.8 labels/Tsc1-D using $\varepsilon_{595} = 104{,}000$ M$^{-1}$ cm$^{-1}$ for Texas Red and a correction factor of $0.26 \times \varepsilon_{595}$ to account for its absorbance at 280 nm. Hsp90$\alpha$ at the indicated concentrations was incubated on ice in 50 mM Tris pH 7.2, 150 mM NaCl, 1 mM TCEP, 4 mM MgCl$_2$ with or without 1 mM ATP, ADP, AMPPNP or 1 $\mu$M GB for 10 min and then incubated with 1 $\mu$M labeled Tsc1-D for 30 min. Fluorescence anisotropy was measured using a SpectraMax i3 equipped with rhodamine fluorescence polarization module ($\lambda_{ex}\backslash\lambda_{em}$ = 535 nm/ 595 nm) (Molecular Devices, Sunnyvale, CA). Curve fitting was done in SigmaPlot 10.0.

## Hsp90 ATPase activity measurement

Purified Hsp90 protein was quantified by Micro BCA™ Protein Assay Kit (Thermo Scientific) and 2.5 $\mu$g Hsp90$\alpha$ was assayed in triplicate as outlined in the P$_i$Per™ Phosphate Assay Kit (Life Technologies) and previously described (Dunn *et al*, 2015). ATP turnover was calculated as mmol P$_i$ per mol Hsp90$\alpha$ per minute, and relative ATPase activity was calculated from those values, with the value of Hsp90$\alpha$ alone representing 100% activity.

## Quantification and statistical analysis

The data presented are the representative or examples of three biological replicates unless it is specified. Data were analyzed with unpaired *t*-test. Asterisks in figures indicate significant differences (\*$P < 0.05$, \*\*$P < 0.005$, \*\*\*$P < 0.0005$, and \*\*\*\*$P < 0.0001$). Error bars represent the standard deviation (SD) for three independent experiments, unless it is indicated.

Expanded View for this article is available online.

## Acknowledgments
We are indebted to Dr. David J. Kwiatkowski (Brigham and Women's Hospital) and Dr Brendan D. Manning (Harvard School of Public Health) for their comments and advice on the manuscript. We are grateful to Dr. Kwiatkowski for Tsc1$^{-/-}$ MEF cell line, Dr. Stephen Goff (Columbia University) for c-Abl$^{-/-}$ MEF cell line, Dr. Brodsky (University of Pittsburgh) for CFTR plasmid, Dr. Timothy Haystead (Duke University) for SNX-2112, and Dr. Jason Gestwicki for JG-98 (University of California San Francisco). This work was supported by the National Institute of General Medical Sciences of the National Institutes of Health under Award Number R01GM124256 (M.M.). The content is solely the responsibility of the authors and does not necessarily represent the official views of the National Institutes of Health. This work was also supported by the Wellcome Trust 095605/Z11/Z (C.P.), SUNY Upstate Medical University, Upstate Foundation, One Square Mile of Hope Foundation, Carol M. Baldwin Breast Cancer Fund (D.B., M.M.), and Urology Care Foundation-American Urological Association (M.M.).

## Author contributions
MRW, RAS, DMD, EM, ARB, RLM, NR, BP, CP, DB, and MM performed experiments. MRW, RAS, DMD, DB, OS, GB, SNL, DHG, MW, and MM designed experiments. MRW, RAS, and MM wrote the manuscript. MM conceived the project.

## Conflict of interest
The authors declare that they have no conflict of interest.

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
