## [Review Process File · The EMBO Journal]

Manuscript EMBO-2017-96700

Tumor suppressor Tsc1 is a new Hsp90 co-chaperone that facilitates folding of kinase and non-kinase clients

Mark R. Woodford, Rebecca A. Sager, Elijah Marris, Diana M. Dunn, Adam R. Blanden, Ryan L. Murphy, Nicholas Rensing, Oleg Shapiro, Barry Panaretou, Chrisostomos Prodromou, Stewart N. Loh, David H. Gutmann, Dimitra Bourboulia, Gennady Bratslavsky, Michael Wong & Mehdi Mollapour

Corresponding author: Mehdi Mollapour, SUNY Upstate Medical University

Review timeline:

Submission date:	08 February 2017
Editorial Decision:	28 March 2017
Revision received:	27 July 2017
Editorial Decision:	21 August 2017
Revision received:	15 September 2017
Accepted:	02 October 2017

Editor: Anne Nielsen

Transaction Report:

1st Editorial Decision

28 March 2017

Thank you for sending a preliminary point-by-point response to the concerns raised by our three referees. I am glad to hear that you find the referee comments constructive and useful, and that you will set up experiments to address most of them. However, we also had to notice that the outcome of the proposed experiments cannot be predicted at the current stage, which could potentially alter the conclusion in the final manuscript.

Based on the input from our referees, I would like to invite you to submit a revised manuscript incorporating the work outlined in your response, provided that the original conclusions still hold true. I should remind you that it is EMBO Journal policy to allow a single round of revision only and that, therefore, acceptance or rejection of the manuscript will depend on the completeness of your responses in this revised version. I do realize that addressing all the referees' criticisms will require a lot of additional time and effort and be technically challenging. I would therefore understand if you wish to publish the manuscript rapidly and without any significant changes elsewhere, in which case please let us know so we can withdraw it from our system. In addition, I want to add that I can understand that you wish to keep an extensive characterization of the implications for mTORC1 activity for a future study but since this is brought up as a major point by the referee, I would encourage you to include at least some data on the effects on downstream signaling.

If you decide to thoroughly revise the manuscript for The EMBO Journal, please include a detailed

point-by-point response to the referees' comments. Please bear in mind that this will form part of the Review Process File, and will therefore be available online to the community. For more details on our Transparent Editorial Process, please visit our website: <http://www.embo.org/embo-press>

We generally allow three months as standard revision time but could offer an extension of up to a total of six months in this case. As a matter of policy, competing manuscripts published during this period will not negatively impact on our assessment of the conceptual advance presented by your study. However, we request that you contact the editor as soon as possible upon publication of any related work, to discuss how to proceed. Should you foresee a problem in meeting this three-month deadline, please let us know in advance and we may be able to grant an extension.

Thank you for the opportunity to consider your work for publication. I look forward to your revision.

REFEREE REPORTS

Referee #1:

In this work, Woodford et al. carry out an exhaustive list of experiments to establish that the tumor suppressor Tsc1 acts as an Hsp90 cofactor that inhibits the chaperone ATPase activity and that competes with another Hsp90 cofactor, Aha1, for the interaction with the chaperone. The authors show that Tsc1 forms a dimer and that its C-terminus interacts with the middle domain of Hsp90. Tsc1 together with Tsc2 form tuberous sclerosis complex (TSC). It is known that Tsc1 stabilizes Tsc2, but the mechanism of this stabilization was not known. Here, the authors suggest that Tsc1 stabilizes Tsc2 as well as other proteins by acting as an Hsp90 cochaperone.

GENERAL COMMENTS

While the manuscript provides very compelling evidence that Tsc1 is a new cochaperone of Hsp90, there is little mechanistic information about how this protein might function. Furthermore, while initially the authors highlight the possibility that Tsc1 acts to stabilize Tsc2 through interactions with Hsp90, this point is not further raised and no specific follow up experiments are done. The manuscript as a whole is more phenotypic and less mechanistic. Finally, several of the experiments need to be better controlled or repeated as discussed below.

SPECIFIC COMMENTS

Tsc2 is a client of Hsp70 and Hsp90

Figure 1A

1. Tsc1 CoIP: why do the authors see two bands instead of one as shown in the inputs?
2. In the article, they claim that Tsc1 and Aha1 compete for Hsp90 binding, but why they do not blot for Aha1 in these experiments?
3. Why is the Hsp70 IP band cutoff?

Figures 1B

1. I understand that in Figure 1C they blot for p-Akt because its phosphorylation is defective in cells lacking Tsc1 and Tsc2, and Akt is an Hsp90 client. However, if Hsp70 inhibition is affecting levels of Tsc2, it should also be affecting levels of p-Akt. Therefore, I would suggest to blot for p-Akt in Figure 1B.

Figure 1C

1. Akt lane: I understand Akt should be the lower band and p-Akt the upper one. I suggest the authors indicate it to avoid misunderstanding. Both p-Akt and Akt are being degraded.
2. Is there a reason why we only see one band at the time 0 but doublets at other times?
3. The authors should also show a blot for Hsp70 and Hsp90. It is important to see what is happening to these chaperones upon inhibition.

Figure 1D

1. p-Akt and Akt lanes look very different from Figure 1C. It is strange to see only one band now for Akt.
2. Why they do not blot for Hsp90 and Hsp70?

Figure 1C-1F

1. The authors should blot for Hsp70 and Hsp90.

Figure 1G

1. What happens to the levels of ubiquitinated Tsc2 when Hsp70 is inhibited? It's known that Hsp70-CHIP can target clients for degradation.

The new co-chaperone Tsc1 inhibits Hsp90 ATPase activity

Figure 2A

1. Co-IP of GR is not convincing.

Figures 2C and 2E

1. Deletion of the MEEVD motif affects Tsc1 binding to Hsp90. However, further experiments showed that Tsc1 binds to the M-domain of Hsp90. The reason for why Tsc1 does not bind to Hsp90 C-domain anymore should be addressed. Is it the FLAG-tag added to Hsp90 N- or C-terminus? If FLAG-tag is at the C-terminus, this might be messing up the interaction. I would suggest performing the experiments of Figure 2C and 2E with Hsp90 N-terminally tagged.

Figure 2F

1. In the presence of GB, the authors claim that Tsc1 does not bind Hsp90. However, there is a positive slope in the binding curve. My question is: did the authors perform a control experiment with some known protein that does not bind Hsp90? Is the behavior similar to Tsc1-Hsp90+GB titration?

Figure 2G

1. I suggest changing the x-axis to Hsp90-Tsc molar ratio instead of Tsc1 concentration. Tsc1 seems to be a potent Hsp90 inhibitor (molar ratio of around 1 Hsp90:3 Tsc1 completely abolishes Hsp90 ATPase activity).

Figure 2H

1. It is very interesting to see that knocking out a co-chaperone impacts the activity of Hsp90 even after purification. How can the authors explain this? Is there any evidence of differential PTMs of Hsp90 when Tsc1 is knocked-out? This question should be addressed.

Tsc1 co-chaperone enables Tsc2 binding to Hsp90

Figure 3A, B and C

1. These are very interesting results. Did the levels of Hsp90 change upon Tsc1 deletion or over-expression? Also, the authors need to blot for Tsc2.

Figure 3D, E and F

1. Very nice result. I understand that Tsc2 stability is dependent on Hsp90. My question is if Tsc2 is binding directly to Hsp90 as a client, in a way that Tsc1 promotes this binding, or if Tsc2 binds to Tsc1, and then Tsc1 binds to Hsp90 (this would be an indirect interaction). I would suggest to perform the same experiments in the presence of GB to discriminate between these 2 mechanisms.

The co-chaperone Tsc1 binds to the closed conformation of Hsp90

Figure 4G and H

1. The fact that Hsp90 D93A binds less to Tsc1, but binds the same to Tsc1-D could suggest a mechanism of two binding sites. How do the authors address this finding?

Tsc1-D dimer binds to both protomers of the Hsp90 middle-domain
Figure 5G

The co-IP in the Figure 5F gives some support to the results of Figure 5G. However, SEC is not the best method to estimate MW of proteins by far, especially if the structure is unknown. Elongated proteins have a different frictional coefficient that affects their elution profile.

Phosphorylation of Aha1 displaces Tsc1 from Hsp90
Figure 6F and G

1. In the DPH untreated cells Tsc1^{+/+} (Figure 6F), Hsp90 interacts strongly with Tsc1 and less with Aha1. In the Figure 6G (GNF5 untreated cells Tsc1^{+/+}), the same control is done but Aha1 is strongly bound to Hsp90 and Tsc1 is not seen. This is very strange. These controls experiments should look the same.

Discussion

Page 16, line 6

The authors argue that Hsp70 and Hsp90 inhibition leads to Tsc2 ubiquitination and degradation. While they have shown the data for Hsp90, they did not show any evidence of ubiquitination when Hsp70 was inhibited.

Page 17, line 3

The authors say a mutation in Tsc1 could destabilize the protein and leads to impairment of Tsc2 binding to Hsp90 resulting in the degradation of Tsc2. It would be important to test one such mutant in this work.

Page 18

It would be very interesting if the authors check the effect of Aha1-Y223E on the maturation of CFTR (as in Figure 3C). It is known that the Aha1-Hsp90 interaction leads to a 'premature' form of CFTR. It would give substantial support to the author's conclusion that the delay in the Hsp90 cycle by Tsc1 binding is responsible in driving proper CFTR maturation.

Referee #2:

This manuscript reports the discovery of a novel Hsp90 co-chaperone, Tsc1.

The molecular chaperone Hsp90 and its numerous cochaperones form an important hub in the protein folding and quality control network of the eukaryotic cell. The cochaperones target specific substrates to Hsp90, control its ATPase cycle and recruit additional activities such as peptidyl-prolyl isomerase activity to the complex.

Mutations in the genes Tsc1 and Tsc2 are the cause of a rare form of cancer, tuberous sclerosis complex (TSC). Tsc2 is a GTPase activating protein. The cancer mutations seem to destabilize Tsc2 and upregulate the mTOR pathway. The 140 kDa protein Tsc1 functions by stabilizing Tsc2 against degradation.

The authors present a plethora of data in support of Tsc1 being a general Hsp90 cochaperone and Tsc2 being a Hsp90 substrate:

First, they show that Tsc2 associates with Tsc1 and the chaperones Hsp70 and Hsp90. Moreover, Tsc2-IP pulls down the Hsp90 cochaperones HOP, PP5, Cdc37 and p23. Inhibition with Hsp70 and Hsp90 inhibitors destabilizes Tsc2; proteasome inhibition stabilizes Tsc2. This suggests that Hsp70 and Hsp90 are involved in quality control of Tsc2. Tsc1 is unaffected, consistent with a cochaperone function. Experiments with purified Tsc2 and Hsp90 suggest that Tsc1 is needed for efficient loading of Tsc2 onto Hsp90.

Tsc1 also associates with Hsp70 and Hsp90, and with typical Hsp90 client proteins, such as the kinases Raf-1, Akt1 and Cdk4 and the steroid receptor GR. Intriguingly, several Hsp90 cochaperones influencing the ATPase cycle of Hsp90, such as HOP, p23 and Aha1 did not associate with Tsc1, suggesting mutually exclusive binding to Hsp90. Tsc1 associates with the middle and C-terminal domains of Hsp90; binding is dependent on the C-terminal MEEVD motif of Hsp90, although Tsc1 does not seem to contain a TPR domain, which is known to associate with MEEVD motifs. The C-terminal 166 residues of Tsc1, Tsc1-D, appear sufficient for Hsp90 binding; this part

exclusively interacts with the middle domain. Tsc1-D inhibits Hsp90 ATPase activity in a dose-dependent manner.

The authors also investigate the interplay of Tsc1 with the ATPase-activating Hsp90 co-chaperone Aha1. Binding of the two factors to the middle domain seems mutually exclusive, and dependent on phosphorylation of Hsp90 at position 313. Binding of Tsc1 and Aha1 is sensitive to a mutation in the middle domain of Hsp90.

In yeast, expression of Tsc1 is toxic, when endogenous Hsp90 is replaced by human Hsp90. Co-expression of a phosphomimetic mutant of human Aha1 relieves toxicity.

Critique:

The amount of data in the manuscript is overwhelming. I am not sure, whether everything that is shown in the main text is needed to demonstrate that Tsc1 functions as a Hsp90 cochaperone. To convey the story, it would be advantageous to explain the actual function of Tsc1 in cells better. Is it highly specialized in chaperoning Tsc2, or a general factor that affects every Hsp90 client protein? Is its cellular abundance similar to Aha1? Is Tsc1 conserved in protists, animals and plants? Why is Tsc1 such a large protein, when the C-terminal snippet is sufficient to regulate Hsp90 ATPase activity? These aspects should be explained in the Introduction and Discussion sections of the manuscript.

Deletion of Tsc1 in vivo appears to destabilize Hsp90 client proteins, in support of the conclusion that Tsc1 is a Hsp90 cochaperone. In order to show that this is a direct consequence of Hsp90 misregulation, the same effects should be shown for Hsp90 inhibition.

Perhaps one could take advantage of the Tsc1 knockout cell lines and express the minimum construct Tsc1-D, in order to investigate the effect on different Hsp90 client proteins. This could show which clients are directly affected by Tsc1 expression, in contrast to indirect effects by unbalancing the competition between Tsc1 and cochaperones Aha1, HOP and p23.

In summary, this is an interesting study. Additional work is mainly needed to make the narrative more digestible.

Referee #3:

The paper by Woodford et al delineates a physical and functional interaction between the Tuberous Sclerosis Complex (TSC) 1 and the Hsp90 chaperone. Using biochemical and functional data, the authors propose that TSC1 functions as an inhibitor of the ATPase activity of Hsp90. In turn, binding of TSC1 to Hsp90 affects cellular levels of several Hsp90 substrates, including kinases involved in signal transduction. These data imply an unexpected connection between the TSC protein complex, traditionally associated with mTOR regulation, and a broad range of physiological processes controlled by Hsp90 and its co-chaperones.

Despite its potential interest, the manuscript suffers from major conceptual weaknesses.

- 1- The exact role of TSC1 toward Hsp90 remains unclear. In Fig 3A and 3B, both TSC1 deletion and its overexpression lead to reduced levels of Hsp90 clients such as ErbB2 and Raf1, which is difficult to reconcile with the biochemical data of Fig 4-6. Although the biochemical data suggest inhibition of Hsp90 catalytic activity by TSC1, whether this translates into genuine regulatory actions is left wide open.
- 2- A crucial point is that the effects of TSC1 deletion and overexpression on cellular levels of TSC2 are not shown. If TSC2 was a client of Hsp90 (as suggested by the results in Fig 1), one would expect that TSC1 deletion/overexpression should have major effects on TSC2 stability. It is surprising that the authors did not check this point.
- 3- The fact that TSC1 is upstream of Hsp90, whereas TSC2 is downstream of it, implies that deletion of TSC1 and TSC2 should not phenocopy each other, which runs contrary to most reports. Have the authors looked at the levels and activity of Erk1, Raf-1, ULK1 and c-Src, side-by-side in TSC1 vs TSC2 null cells?
- 4- Does the TSC1-Hsp90 connection play a role in mTORC1 signaling or does it represent a completely separate activity? The authors should perform a basic characterization of mTORC1 signaling in TSC1 vs TSC2 null cells. They should also rescue TSC1 null cells with TSC1 constructs that can or cannot bind to Hsp90 and determine their ability to restore mTORC1 regulation.
- 5- The fact that TSC1 and TSC2 interact with both Hsp90 and Hsp70 raises concerns that the

interaction may not be as specific as the authors propose. Since the Hsp70 interaction is shown in the paper, its reproducibility and significance should be documented, at least in part.

Additional points.

- 1- The interaction between TSC2 and Hsp90 (Fig 1A) lacks adequate negative controls. IgG is not sufficient, another protein of comparable size to TSC2 should be provided.
- 2- Similarly, in Fig 2A, an adequate negative control must be provided for the interaction of TSC1 with Hsp90.
- 3- The ATPase assay in Fig 2G also lacks a negative control protein, therefore it cannot be concluded that TSC1 (D-domain) is a specific Hsp90 inhibitor.

1st Revision - authors' response

27 July 2017

Dr Anne Nielsen
Editor
The EMBO Journal
Postfach 1022.40,
69012 Heidelberg,
Germany

UPSTATE
MEDICAL UNIVERSITY

Department of Urology
*Department of Biochemistry &
Molecular Biology*

750 East Adams St
Syracuse, NY 13210
Phone: +1-315-464-8749
Fax +1-315-464-8750
mollapom@upstate.edu
www.mollapourlab.com

July 27, 2017

RE: EMBOJ-2017-96700R

Dear Dr Nielsen,

Thank you for giving us the opportunity to address the reviewer's questions and concerns. We are also grateful to the reviewers for their constructive and insightful comments on our manuscript by Woodford et al., entitled "Tsc1 co-chaperone breaks the asymmetric binding of Aha1 with Hsp90 and inhibits the ATPase activity". We are happy to see that the reviewers found our study interesting and significant.

We were able to carry out all our proposed experiments in our original response to the reviewer's comments except for one. Unfortunately we were unable to silence *AHA1* (activating co-chaperone) by siRNA in Tsc1^{-/-} MEF cells and examine the stability and activity of the clients (eg ErbB2, Raf1, ULK1, c-Src, Cdk4, GR, FLCN and Tsc2) by immunoblotting. We believe this was due to abnormalities associated with the growth of Tsc1^{-/-} MEF cells.

In the revised manuscript, we have incorporated additional and revised data in Fig 1A, B, C, D, E, F, G; Fig 2A; Fig 3A, B, C; Fig 6F, G. New results are presented in Fig 3G, H, I, J, K; Fig EV1, EV2B, E; EV3.

Below, we provide a detailed point-by-point response to the reviewers' questions and explanation of how the new additions and revisions address their concerns. The new data has only strengthened our original claim.

We would also like to point out that we have included 4 additional figures (referred to as Response Fig) in the response below that are not presented in the manuscript. We strongly believe that these results are too preliminary and also not within the scope of our manuscript.

Thank you for considering our manuscript for publication in *The EMBO Journal*.

Sincerely,

Mehdi Mollapour PhD
Assistant Professor of Urology
Head, Renal Cancer Biology Section
Department of Urology
SUNY Upstate Medical University

We are grateful to the reviewers for their constructive and insightful comments on our manuscript. We are happy to see that the reviewers found our study interesting and significant.

In the revised manuscript, we have incorporated additional and revised data in Fig 1A, B, C, D, E, F, G; Fig 2A; Fig 3A, B, C; Fig 6F, G.

New results are presented in Fig 3G, H, I, J, K; Fig EV1, EV2B, E; EV3. These new data did not exist in the original manuscript.

Below, we provide a detailed point-by-point response to the reviewers' questions and explanation of how the new additions and revisions address their concerns. We also believe that the new data has strengthened our original claim.

We would also like to point out that we have included 4 additional figures (referred to as Response Fig) in the response below that are not presented in the manuscript. Although these results are informative, we strongly believe that they are too preliminary and not within the scope of our manuscript.

Referee #1:

In this work, Woodford et al. carry out an exhaustive list of experiments to establish that the tumor suppressor Tsc1 acts as an Hsp90 cofactor that inhibits the chaperone ATPase activity and that competes with another Hsp90 cofactor, Aha1, for the interaction with the chaperone. The authors show that Tsc1 forms a dimer and that its C-terminus interacts with the middle domain of Hsp90. Tsc1 together with Tsc2 form tuberous sclerosis complex (TSC). It is known that Tsc1 stabilizes Tsc2, but the mechanism of this stabilization was not known. Here, the authors suggest that Tsc1 stabilizes Tsc2 as well as other proteins by acting as an Hsp90 cochaperone.

GENERAL COMMENTS

While the manuscript provides very compelling evidence that Tsc1 is a new cochaperone of Hsp90, there is little mechanistic information about how this protein might function. Furthermore, while initially the authors highlight the possibility that Tsc1 acts to stabilize Tsc2 through interactions with Hsp90, this point is not further raised and no specific follow up experiments are done. The manuscript as a whole is more phenotypic and less mechanistic. Finally, several of the experiments need to be better controlled or repeated as discussed below.

We would like to thank the reviewer for their constructive and insightful comments to our manuscript. We also appreciate their positive note on our findings, however we would like to respond to their comment on "*little mechanistic information about how this protein might function*" and "*the manuscript as a whole is more phenotypic and less mechanistic.*"

We have identified Tsc1 as a new co-chaperone and Tsc2 as a new client of Hsp90. Tsc1 decelerates the chaperone cycle by inhibiting Hsp90 ATPase-activity, therefore loading Tsc2 on Hsp90. This prevents Tsc2 from ubiquitination and proteasomal degradation. Our work has also mechanistically demonstrated that Tsc1 carboxy-domain homodimerizes and binds to both protomers of the Hsp90 middle-domain, which is sufficient to inhibit Hsp90 ATPase activity. Tsc1 interacts with the Aha1 binding sites on Hsp90 middle-domain, therefore breaking the asymmetric interaction of Aha1 with Hsp90. Our data has also led us to speculate that Tsc1 prevents the conformational changes in the catalytic loop of Hsp90 and stops the release of R400 from its retracted inactivating conformation consequently inhibiting Hsp90 ATPase activity. The latter statement is only a speculation on how exactly Tsc1 might inhibit Hsp90 ATPase activity, which requires structural data in order to confirm this phenomenon. This is clearly not within the scope of this paper.

SPECIFIC COMMENTS

Tsc2 is a client of Hsp70 and Hsp90

Figure 1A

1. Tsc1 CoIP: why do the authors see two bands instead of one as shown in the inputs?

We appreciate the reviewer's comment and we have presented an improved blot showing a single band for Tsc1 co-IP.

2. In the article, they claim that Tsc1 and Aha1 compete for Hsp90 binding, but why they do not blot for Aha1 in these experiments?

We have probed for Aha1 in the blot for Fig 2A and the data was presented in Fig EV2A. We have already presented data in Fig 5A, B, C, D, E showing the binding of Tsc1 and Aha1 to Hsp90. Fig5H and Fig5I show the dynamic of Aha1 and Tsc1-D fragment interaction with Hsp90.

3. Why is the Hsp70 IP band cutoff?

We are presenting a better blot for Hsp70.

Figures 1B

1. I understand that in Figure 1C they blot for p-Akt because its phosphorylation is defective in cells lacking Tsc1 and Tsc2, and Akt is an Hsp90 client. However, if Hsp70 inhibition is affecting levels of Tsc2, it should also be affecting levels of p-Akt. Therefore, I would suggest to blot for p-Akt in Figure 1B.

We have provided new data in the revised Figure 1B. As the reviewer correctly suggested, the level of p-Akt has significantly decreased upon Hsp90 inhibition.

Figure 1C

1. Akt lane: I understand Akt should be the lower band and p-Akt the upper one. I suggest the authors indicate it to avoid misunderstanding. Both p-Akt and Akt are being degraded.

2. Is there a reason why we only see one band at the time 0 but doublets at other times?

3. The authors should also show a blot for Hsp70 and Hsp90. It is important to see what is happening to these chaperones upon inhibition.

We thank the reviewer and we have replaced the p-Akt and Akt blots.

Blots for Hsp70 and Hsp90 have also been included.

Figure 1D

1. p-Akt and Akt lanes look very different from Figure 1C. It is strange to see only one band now for Akt.

We have replaced the blots in Figure 1C. This is consistent with our previous data in Figure 1B.

2. Why they do not blot for Hsp90 and Hsp70?

Blots for Hsp70 and Hsp90 have been added.

Figure 1C-1F

1. The authors should blot for Hsp70 and Hsp90.

Hsp70 and Hsp90 blots have been included.

Figure 1G

1. What happens to the levels of ubiquitinated Tsc2 when Hsp70 is inhibited? Its known that Hsp70-CHIP can target clients for degradation.

We have provided new data showing that inhibition of Hsp70 leads to ubiquitination of Tsc2.

The new co-chaperone Tsc1 inhibits Hsp90 ATPase activity

Figure 2A

1. Co-IP of GR is not convincing.

We have re-run the samples and have included a new blot for the Co-IP of GR.

Figures 2C and 2E

1. Deletion of the MEEVD motif affects Tsc1 binding to Hsp90. However, further experiments showed that Tsc1 binds to the M-domain of Hsp90. The reason for why Tsc1 does not bind to Hsp90 C-domain anymore should be addressed. Is it the FLAG-tag added to Hsp90 N- or C-terminus? If FLAG-tag is at the C-terminus, this might be messing up the interaction. I would suggest performing the experiments of Figure 2C and 2E with Hsp90 N-terminally tagged.

We thank the reviewer and we would like to explain that Hsp90 constructs in Figures 2C and 2E are FLAG-tagged at the N-domain.

Our finding is not unusual and that is why we included HOP as a control. Elegant work by Agard's group using cryo-EM structure has shown the Hsp90 MD-CTD junction to be the primary binding interaction between Hsp90 and HOP (Southworth & Agard, 2011). Also, work by Buchner's group (Schmid et al., 2012) has shown that Sti1/HOP binds to both MEEVD and the middle-domain of Hsp90.

It is perhaps possible that Tsc1 employs a similar two-step mechanism for binding to Hsp90. However, our data suggest that the Tsc1-D fragment binding to the middle domain of Hsp90 is sufficient to inhibit its ATPase activity. Again, as it was mentioned above, this requires structural data, which is not within the scope of this paper.

Figure 2F

1. In the presence of GB, the authors claim that Tsc1 does not bind Hsp90. However, there is a positive slope in the binding curve. My question is: did the authors perform a control experiment with some known protein that does not bind Hsp90? Is the behavior similar to Tsc1-Hsp90+GB titration?

This is a very interesting comment by the reviewer. We have carried out the binding assay with labeled BSA showing that it does not interact with Hsp90 (Fig EV2B). The binding curve of Tsc1-Hsp90+GB suggests that Tsc1-Hsp90 specific interaction is significantly reduced compared to non-drug control. We will therefore modify our claim from "*lack of Tsc1 binding to Hsp90 in the presence of GB*" to "*Tsc1-D-His6 binding to Hsp90 is significantly reduced in the presence of GB*".

Figure 2G

1. I suggest changing the x-axis to Hsp90-Tsc molar ratio instead of Tsc1 concentration. Tsc1 seems to be a potent Hsp90 inhibitor (molar ratio of around 1 Hsp90:3 Tsc1 completely abolishes Hsp90 ATPase activity).

We thank the reviewer for this constructive comment. We have changed the x-axis to molar ratio.

Figure 2H

1. It is very interesting to see that knocking out a co-chaperone impacts the activity of Hsp90 even after purification. How can the authors explain this? Is there any evidence of differential PTMs of Hsp90 when Tsc1 is knocked-out? This question should be addressed.

(Text related to Response Fig. 1 for referees not shown)

A
A
A
A
A
A

Tsc1 co-chaperone enables Tsc2 binding to Hsp90

Figure 3A, B and C

1. These are very interesting results. Did the levels of Hsp90 change upon Tsc1 deletion or over-expression? Also, the authors need to blot for Tsc2.

We have probed for Tsc2 and Hsp90 in Fig 3A, B and C. The levels of Hsp90 are unaffected by Tsc1 deletion or over-expression.

Figure 3D, E and F

1. Very nice result. I understand that Tsc2 stability is dependent on Hsp90. My question is if Tsc2 is binding directly to Hsp90 as a client, in a way that Tsc1 promotes this binding, or if Tsc2 binds to Tsc1, and then Tsc1 binds to Hsp90 (this would be an indirect interaction). I would suggest to perform the same experiments in the presence of GB to discriminate between these 2 mechanisms.

We thank the reviewer for their kind words and their great suggestion.

We carried out this experiment and our data showed that treating the Tsc1:Hsp90 complex with GB does not disrupt Hsp90 interaction with Tsc1 (Fig 3G). The interaction of Tsc1:Tsc2 is also not disrupted with GB (Fig 3H). We next formed a trimeric Tsc1:Tsc2:Hsp90 complex and then treated the samples with GB showing the dissociation of Tsc2 from the complex (Fig 3I). However, Tsc1:Hsp90 interaction was unaffected (Fig 3J), suggesting that Tsc1 act as a loading scaffold facilitating the direct binding of Tsc2 to Hsp90 (Fig 3K).

The co-chaperone Tsc1 binds to the closed conformation of Hsp90

Figure 4G and H

1. The fact that Hsp90 D93A binds less to Tsc1, but binds the same to Tsc1-D could suggest a mechanism of two binding sites. How do the authors address this finding?

We completely agree with the reviewer and that is why we chose our comments very carefully. In the Discussion we wrote; *“Our data suggest that the binding of Tsc1 to Hsp90 is complex, and it requires multiple surfaces of Tsc1 to interact with Hsp90.”*

We did not think it would be useful or appropriate to make these comments at the end of the *“The co-chaperone Tsc1 binds to the closed conformation of Hsp90”* in the Result section.

Tsc1-D dimer binds to both protomers of the Hsp90 middle-domain

Figure 5G

The co-IP in the Figure 5F gives some support to the results of Figure 5G. However, SEC is not the best method to estimate MW of proteins by far, especially if the structure is unknown. Elongated proteins have a different frictional coefficient that affects their elution profile.

We performed gel filtration on the purified protein. Tsc1-D-His₆ migrated as a monodisperse peak with an apparent molecular weight of approximately twice the predicted molecular weight of the monomer (22.1 kDa). Of course this data alone cannot confirm the dimerization of Tsc1-D. However, our results (Fig 5F and G) collectively indicate that the protein is a dimer in solution.

Phosphorylation of Aha1 displaces Tsc1 from Hsp90
Figure 6F and G

1. In the DPH untreated cells Tsc1^{+/+} (Figure 6F), Hsp90 interacts strongly with Tsc1 and less with Aha1. In the Figure 6G (GNF5 untreated cells Tsc1^{+/+}), the same control is done but Aha1 is strongly bound to Hsp90 and Tsc1 is not seen. This is very strange. These controls experiments should look the same.

We thank the reviewer for their constructive comment. The reason for this inconsistency is because of different exposures of the blots. We have replaced the Co-IP of Tsc1 in Fig 6G to reflect a more comparative exposure to Fig 6F.

Discussion

Page 16, line 6

The authors argue that Hsp70 and Hsp90 inhibition leads to Tsc2 ubiquitination and degradation. While they have shown the data for Hsp90, they did not show any evidence of ubiquitination when Hsp70 was inhibited.

We have included additional data with the Hsp70 inhibitor. Please see our response to the initial comment for Fig 1G.

Page 17, line 3

The authors say a mutation in Tsc1 could destabilize the protein and leads to impairment of Tsc2 binding to Hsp90 resulting in the degradation of Tsc2. It would be important to test one such mutant in this work.

It has previously been shown by (Hoogeveen-Westerveld et al., 2011) that mutation of Tsc1-L117P destabilizes Tsc1. Based on previous studies, this mutation is not within the Tsc2 binding domain, and thus when stable should be able to bind Tsc2. We have created this *Tsc1-L117P* mutant and transiently expressed different amounts of the construct in TSC1^{-/-} MEF cells (Fig EV3A). Our data showed that overexpression of Tsc1-L117P correlates to its stabilization as well as stability of Tsc2 (Fig EV3A). Furthermore, expression of 1µg Tsc1-L117P in TSC1^{-/-} MEF cells is undetectable by Western blot, however treatment of these cells with the proteasome inhibitor bortezomib leads to stabilization of Tsc1-L117P (Fig EV3B). Finally, we overexpressed Tsc1-L117P and could detect it by Western blot (Fig EV3C). Immunoprecipitation of this mutant allowed us to Co-IP and detect both Tsc2 and Hsp90 together (Fig EV3C). We believe our data at least partially addresses the reviewer's comment.

Page 18

It would be very interesting if the authors check the effect of Aha1-Y223E on the maturation of CFTR (as in Figure 3C). It is known that the Aha1-Hsp90 interaction leads to a 'premature' form of CFTR. It would give substantial support to the author's conclusion that the delay in the Hsp90 cycle by Tsc1 binding is responsible in driving proper CFTR maturation.

We appreciate this great comment by the reviewer. We have previously reported the effect of Aha1-Y223E on the maturation of CFTR (Response Fig 2), (Dunn et al., 2015).

Please see below.

Response Fig 2. HEK293 cells were co-transfected with CFTR and hAha1-FLAG (WT) and Y223F and Y223E mutants. After 24 hr, CFTR, hAha1-FLAG, and hHsp90 were detected by immunoblotting. Taken from (Dunn et al., 2015).

Referee #2:

This manuscript reports the discovery of a novel Hsp90 co-chaperone, Tsc1.

The molecular chaperone Hsp90 and its numerous cochaperones form an important hub in the protein folding and quality control network of the eukaryotic cell. The cochaperones target specific substrates to Hsp90, control its ATPase cycle and recruit additional activities such as peptidyl-prolyl isomerase activity to the complex.

Mutations in the genes Tsc1 and Tsc2 are the cause of a rare form of cancer, tuberous sclerosis complex (TSC). Tsc2 is a GTPase activating protein. The cancer mutations seem to destabilize Tsc2 and upregulate the mTOR pathway. The 140 kDa protein Tsc1 functions by stabilizing Tsc2 against degradation.

The authors present a plethora of data in support of Tsc1 being a general Hsp90 cochaperone and Tsc2 being a Hsp90 substrate:

First, they show that Tsc2 associates with Tsc1 and the chaperones Hsp70 and Hsp90. Moreover, Tsc2-IP pulls down the Hsp90 cochaperones HOP, PP5, Cdc37 and p23. Inhibition with Hsp70 and Hsp90 inhibitors destabilizes Tsc2; proteasome inhibition stabilizes Tsc2. This suggests that Hsp70 and Hsp90 are involved in quality control of Tsc2. Tsc1 is unaffected, consistent with a cochaperone function. Experiments with purified Tsc2 and Hsp90 suggest that Tsc1 is needed for efficient loading of Tsc2 onto Hsp90.

Tsc1 also associates with Hsp70 and Hsp90, and with typical Hsp90 client proteins, such as the kinases Raf-1, Akt1 and Cdk4 and the steroid receptor GR. Intriguingly, several Hsp90 cochaperones influencing the ATPase cycle of Hsp90, such as HOP, p23 and Aha1 did not associate with Tsc1, suggesting mutually exclusive binding to Hsp90. Tsc1 associates with the middle and C-terminal domains of Hsp90; binding is dependent on the C-terminal MEEVD motif of Hsp90, although Tsc1 does not seem to contain a TPR domain, which is known to associate with MEEVD motifs. The C-terminal 166 residues of Tsc1, Tsc1-D, appear sufficient for Hsp90 binding; this part exclusively interacts with the middle domain. Tsc1-D inhibits Hsp90 ATPase activity in a dose-dependent manner.

The authors also investigate the interplay of Tsc1 with the ATPase-activating Hsp90 co-chaperone Aha1. Binding of the two factors to the middle domain seems mutually exclusive, and dependent on phosphorylation of Hsp90 at position 313. Binding of Tsc1 and Aha1 is sensitive to a mutation in the middle domain of Hsp90. In yeast, expression of Tsc1 is toxic, when endogenous Hsp90 is replaced by human Hsp90. Co-expression of a phosphomimetic mutant of human Aha1 relieves toxicity.

In summary, this is an interesting study. Additional work is mainly needed to make the narrative more digestible.

Critique:

The amount of data in the manuscript is overwhelming. I am not sure, whether everything that is shown in the main text is needed to demonstrate that Tsc1 functions as a Hsp90 cochaperone.

To convey the story, it would be advantageous to explain the actual function of Tsc1 in cells better. Is it highly specialized in chaperoning Tsc2, or a general factor that affects every Hsp90 client protein?

We appreciate the reviewer's comment and as we have already mentioned in the Introduction and Discussion, the current literature describe the role of Tsc1 in the stability of Tsc2, hence down-regulation of the mTORC1 pathway (Crino et al., 2006, Neuman & Henske, 2011). However, our findings suggest an additional function, mainly regulation of Hsp90 in cells. Our results (Fig 2I, 3A, 3B, 3C) suggest that Tsc1 is involved in not only chaperoning of Tsc2 but also other Hsp90 clients (ie kinases and non-kinases). We have explicitly stated this in the Discussion; "*Tsc1 is a new co-chaperone of Hsp90 and our findings suggest that the newly identified function of Tsc1 is to regulate the chaperoning of Hsp90 clients including Tsc2*". We have explained this point in detail in the Discussion section.

Is its cellular abundance similar to Aha1?

This is a very good question and based on the published data (Carpy et al., 2014) available on the fission yeast genome database (<https://www.pombase.org/>), *S. pombe* has approximately 400,000 Aha1 molecules/cell; 6,200 Tsc1 molecules/cell and 883,920 Hsp90 molecules/cell Hsp90. However, as it has been demonstrated previously, PTMs of Hsp90 and its co-chaperones can dramatically impact the binding affinity (Dunn et al., 2015, Mollapour et al., 2010, Mollapour et al., 2011, Soroka et al., 2012).

Is Tsc1 conserved in protists, animals and plants?

Plants appear to lack TSC1/2 complex altogether (Diaz-Troya et al., 2008, Vernoud et al., 2003) despite the presence of TORC1 (Deprost et al., 2007, Menand et al., 2002).

Tsc1 orthologs have been found in animals and fungi but not in *C. elegans* and *S. cerevisiae*. Furthermore, while Tsc1 orthologs are always observed together with Tsc2 orthologs in the same genomes, Tsc2 can be found on its own in additional eukaryotic species (see below figure from (van Dam et al., 2011)). Interestingly GAP domain exists in Tsc2 from these species (*D. discoideum*, *C. merolae*, *P. infestans*, *P. sojae*, *Phaeodactylum tricornutum*) but not the Tsc1-binding domain that is necessary for interaction with Tsc1. Additionally the existence of Tsc1 orthologs in animals and fungi is similar to phylogenetic distribution of the Tsc1-binding domain of Tsc2. This suggests that Tsc2 stability depends on another mechanism that may be independent of Hsp90 or perhaps Tsc2 takes advantage of another co-chaperone for its' binding to Hsp90. We will definitely explore these possibilities in our future research.

Fig. 1 Absence/presence plots in a subset of 65 eukaryotic genomes. Animals and fungi have both TORC1 and TORC2 while plants have only TORC1 and ciliates have only TORC2. Apparently it is possible to lose either one of the complexes while maintaining the other. The GAP domain of TSC2 is well conserved and is found (with few exceptions) in species that also contain Rheb throughout the eukaryotic lineages. TSC1 is an animal/fungal invention and therefore newer than TSC2 and Rheb. The occurrence of TCTP in species lacking Rheb and vice versa, raises additional doubt on the debated Guanine Exchange Factor function of TCTP

(Taken from (van Dam et al., 2011))

Why is Tsc1 such a large protein, when the C-terminal snippet is sufficient to regulate Hsp90 ATPase activity? These aspects should be explained in the Introduction and Discussion sections of the manuscript.

We appreciate this comment and we have discussed this point in the Discussion and very briefly in Introduction section because of character limits set by the journal.

More specifically, although Tsc1-D fragment is sufficient to inhibit Hsp90 ATPase activity, other regions of Tsc1, as it has been shown previously, are essential for binding to Tsc2. We expect that Tsc1's role as a co-chaperone is dependent not just on its effect on Hsp90 ATPase activity but possibly also on other functions such as loading clients to Hsp90. It is worth mentioning that we have also recently identified and reported new co-chaperones of Hsp90, FNIP1 (131 kDa) and FNIP2 (125 kDa), which are similar size in molecular weight to Tsc1 (130kDa) (Woodford et al., 2016a).

Deletion of Tsc1 in vivo appears to destabilize Hsp90 client proteins, in support of the conclusion that Tsc1 is a Hsp90 cochaperone. In order to show that this is a direct consequence of Hsp90 mis-regulation, the same effects should be shown for Hsp90 inhibition.

We appreciate the reviewer's comment, however we respectfully oppose inclusion of these data, because we have carefully chosen these clients based on previous publications, including some from our group (Dunn et al., 2015, Woodford et al., 2016a, Woodford et al., 2016b). They have been characterized and classified as the *bona fide* clients of Hsp90. Although the suggested experiment could serve as an additional control, our "*overwhelming amount of data*" contains adequate positive and negative controls that inclusion of data with Hsp90 inhibitors would appear redundant.

Perhaps one could take advantage of the Tsc1 knockout cell lines and express the minimum construct Tsc1-D, in order to investigate the effect on different Hsp90 client proteins. This could show which clients are directly affected by Tsc1 expression, in contrast to indirect effects by unbalancing the competition between Tsc1 and cochaperones Aha1, HOP and p23.

This is a great suggestion by the reviewer. In response, we have carried out the following experiments;

A) Transiently over-expressed Tsc1-D fragment in Tsc1^{-/-} MEF cells and examined the stability and activity of the clients (eg ErbB2, Raf1, ULK1, c-Src, Cdk4, GR, FLCN and Tsc2) by immunoblotting.

Our data showed that overexpression of Tsc1-D-FLAG in Tsc1^{-/-} MEF cells led to marked increase in c-Src activation (as observed by phospho-Y416-c-Src). In contrast, GR levels were slightly reduced. Further, the levels of Tsc2 were unchanged (see below, Response Fig 3A).

B) Transiently over-expressed Tsc1-D fragment in HEK293 cells and then evaluated the stability and activity of the clients (eg ErbB2, Raf1, ULK1, c-Src, Cdk4, GR, FLCN and Tsc2) by immunoblotting (Response Fig 3B).

Again we found activation of c-Src as assessed by phospho-Y416-c-Src antibody, and surprisingly, a reduction in Tsc2 stability. See figure below (Response Fig 3B).

Taken together, these data suggest that overexpression of Tsc1-D in either the presence or absence of endogenous Tsc1 leads to activation of c-Src (see below, Response Fig 3A, B). We speculate that this effect is through deceleration of Hsp90 ATPase activity or perhaps as the reviewer rightly suggested, the consequence of imbalance of interaction of other co-chaperones such as Aha1, HOP or p23 with Hsp90. Additionally, our data suggest that the co-chaperone function of Tsc1 towards the other clients requires additional domains of Tsc1. Since Tsc1-D lacks these domains and can no longer bind to Tsc2, it therefore cannot enhance the stability of Tsc2 in the Tsc1^{-/-} MEF cells. Our surprising observation of Tsc2 destabilization in HEK293 cells overexpressing Tsc1-D fragment could be the result of its competition with endogenous Tsc1 for binding to Hsp90 (Response Fig 3B). Dissecting the dynamic of the Tsc1-D fragment binding to the Hsp90 co-chaperone complex requires additional

experiments and we plan to address these questions in our future work. Therefore we have decided not to include these data in the current manuscript.

(Response Fig 3. for referees not shown)

Referee #3:

The paper by Woodford et al delineates a physical and functional interaction between the Tuberous Sclerosis Complex (TSC) 1 and the Hsp90 chaperone. Using biochemical and functional data, the authors propose that TSC1 functions as an inhibitor of the ATPase activity of Hsp90. In turn, binding of TSC1 to Hsp90 affects cellular levels of several Hsp90 substrates, including kinases involved in signal transduction. These data imply an unexpected connection between the TSC protein complex, traditionally associated with mTOR regulation, and a broad range of physiological processes controlled by Hsp90 and its co-chaperones.

Despite its potential interest, the manuscript suffers from major conceptual weaknesses.

We are grateful to the reviewer for the above summary of our manuscript. However, we would like to clarify a few points that the reviewer may have misunderstood in our manuscript. Previous work has shown that Tsc1 protects Tsc2 from interacting with HERC1 ubiquitin ligase and prevents its ubiquitin-mediated degradation (Chong-Kopera et al., 2006). However, it is unclear if Tsc1 has a chaperone or a co-chaperone activity towards Tsc2. This has remained one of the major gaps in our knowledge with respect to the TSC research field.

The reviewer is correct by stating that we have used biochemical, biophysical, and functional cell-based assays to conclude that Tsc1 is a new co-chaperone of Hsp90 by inhibiting its ATPase activity. This allows Hsp90 to chaperone its kinase and non-kinase clients including Tsc2, therefore preventing their ubiquitination and proteasomal degradation. Our findings will conceptually advance not only the molecular chaperone research area but also the TSC field, because it has emphasized the importance of Tsc1 function in different signaling pathways through regulating Hsp90 chaperone function.

1- The exact role of TSC1 toward Hsp90 remains unclear. In Fig 3A and 3B, both TSC1 deletion and its overexpression lead to reduced levels of Hsp90 clients such as ErbB2 and Raf1, which is difficult to reconcile with the biochemical data of Fig 4-6. Although the biochemical data suggest inhibition of Hsp90 catalytic activity by TSC1, whether this translates into genuine regulatory actions is left wide open.

We appreciate the reviewer's comment, and we would like to explain that there are common assays agreed upon by the experts in the Hsp90 field to evaluate the function of Hsp90 in cells. It is only when we take these data together that we are able to assess and evaluate Hsp90 chaperone function. These assays involve the binding of Hsp90 to its previously characterized client and co-chaperone proteins, the stability and activity of the clients, and the ability of Hsp90 to bind and hydrolyze ATP.

Data generated from these experiments allows us to evaluate the chaperone activity of Hsp90. This is exactly what we have achieved here and our data has shown that Tsc1 is a decelerator/inhibitor of the Hsp90 chaperone function and it also competes with the activator co-chaperone Aha1 for binding to Hsp90 (Figure 4-6). In Figure 3A the reviewer will notice that deletion of *TSC1* reduces the stability of the kinase (eg ErbB2, Raf1, ULK1, c-Src and CDK4) and non-kinase clients such as GR, Tsc2, and the tumor suppressor FLCN. This is due to hyperactivity of the Hsp90 chaperone cycle and perhaps inability of these clients to be loaded to Hsp90 as the result of *TSC1* absence in cells.

In Figure 3B overexpression of *TSC1* and deceleration/inhibition of Hsp90 chaperone function actually increases the stability of GR and FLCN and also the activity of GR. This is in agreement with the current model in the Hsp90 field that the non-kinase clients such as steroid hormone receptors generally prefer "slow" Hsp90 chaperone function. We further demonstrated this point with another non-kinase client, CFTR, in Figure 3C. Over-expression of *TSC1* actually improved the chaperoning of CFTR.

This effect gets confusing when we examine the kinase clients such as ErbB2, Raf1, ULK1, and c-Src where hyper- or hypoactive Hsp90 as the result of absence or over-expression of *TSC1* can lead to destabilization or down-regulation of these kinase clients. These kinase clients are very sensitive to changes to the Hsp90 chaperone cycle. We have included Cdk4 as a control, because it is already established that a slow chaperone cycle of Hsp90 has very little effect on Cdk4 stability.

To address the reviewer's comment and unravel the "genuine regulatory actions of Tsc1", we have carried out further experiments in order to delineate Tsc1 co-chaperone function as a decelerator/inhibitor of Hsp90 chaperone function and a scaffold co-chaperone that facilitates client binding to Hsp90.

This was achieved by;

A) Transiently over-expressed Tsc1-D fragment in Tsc1^{-/-} MEF cells and examined the stability and activity of the clients (eg ErbB2, Raf1, ULK1, c-Src, Cdk4, GR, FLCN and Tsc2) by immunoblotting.

Our data showed that overexpression of Tsc1-D-FLAG in Tsc1^{-/-} MEF cells led to marked increase in c-Src activation (as observed by phospho-Y416-c-Src). In contrast, GR levels were slightly reduced. Further, the levels of Tsc2 were unchanged (Response Fig 3A).

B) Transiently over-expressed Tsc1-D fragment in HEK293 cells and then evaluated the stability and activity of the clients (eg ErbB2, Raf1, ULK1, c-Src, Cdk4, GR, FLCN and Tsc2) by immunoblotting (Response Fig 3B).

Again we found activation of c-Src as assessed by phospho-Y416-c-Src antibody, and surprisingly, a reduction in Tsc2 stability. See figure below.

Taken together, these data suggest that overexpression of Tsc1-D in either the presence or absence of endogenous Tsc1 leads to activation of c-Src (Response Fig 3A, B). We speculate that this effect is through deceleration of Hsp90 ATPase activity. Additionally, our data suggest that the co-chaperone function of Tsc1 towards the other clients requires additional domains of Tsc1. Since Tsc1-D lacks these domains and can no longer bind to Tsc2, it therefore cannot enhance the stability of Tsc2 in the Tsc1^{-/-} MEF cells. Our surprising observation of Tsc2 destabilization in HEK293 cells overexpressing Tsc1-D fragment could be the result of its competition with endogenous Tsc1 for binding to Hsp90 (Response Fig 3B). Dissecting the dynamic of the Tsc1-D fragment binding to the Hsp90 co-chaperone complex requires additional experiments and we plan to address these questions in our future work. Therefore we have decided not to include these data in the current manuscript.

(Response Fig 3. for referees not shown)

It is noteworthy that these data were also requested by the reviewer #2.

2- A crucial point is that the effects of TSC1 deletion and overexpression on cellular levels of TSC2 are not shown. If TSC2 was a client of Hsp90 (as suggested by the results in Fig 1), one would expect that TSC1 deletion/overexpression should have major effects on TSC2 stability. It is surprising that the authors did not check this point.

We have carried out this experiment. Please see our response to comment 1 above. Additionally we have probed Tsc2 in Fig 3A, B and C, and Fig EV2A, B, in order to show the effects of *TSC1* deletion and overexpression on the stability of Tsc2. Taken together, absence of Tsc1 significantly reduces the stability of Tsc2. Conversely overexpression of Tsc1 had a slight positive impact on the stability of Tsc2.

3- The fact that TSC1 is upstream of Hsp90, whereas TSC2 is downstream of it, implies that deletion of TSC1 and TSC2 should not phenocopy each other, which runs contrary to most reports. Have the authors looked at the levels and activity of Erk1, Raf-1, ULK1 and c-Src, side-by-side in TSC1 vs TSC2 null cells?

We thank the reviewer, however we respectfully disagree with their comment for the following reasons;

Tsc1 ablation results in phenotypes that closely resemble Tsc2 mutants in different organisms (Orlova & Crino, 2010). This, together with the well-established Tsc1-Tsc2 interaction, led to the general view that Tsc1 and Tsc2 act exclusively in complex. Previous works (Miloloza et al., 2002, Thien et al., 2015) challenge the common notion that Tsc1 and Tsc2 are strictly interdependent. Further evidence for separate functions of Tsc1 and Tsc2 comes from microarray and proteomic approaches, which reveal that the *TSC* genes trigger substantially different cellular responses (Hengstschlager et al., 2005, Rosner et al., 2005). Interestingly, in renal and bladder cancers *TSC1* mutations seem to be more prevalent, as compared to *TSC2* (Hornigold et al., 1999, Kucejova et al., 2011, Pymar et al., 2008).

Finally, inactivation of *TSC2* causes a different epilepsy phenotype than *TSC1* inactivation in a mouse model of TSC (Zeng et al., 2011). Due to this body of evidence along with our current manuscript we strongly believe that *TSC1* and *TSC2* deletion do not phenocopy each other. Therefore the proposed experiments would not add any new information to our story presented here.

4- Does the TSC1-Hsp90 connection play a role in mTORC1 signaling or does it represent a completely separate activity? The authors should perform a basic characterization of mTORC1 signaling in TSC1 vs TSC2 null cells. They should also rescue TSC1 null cells with TSC1 constructs that can or cannot bind to Hsp90 and determine their ability to restore mTORC1 regulation.

We are grateful to the reviewer for suggesting these experiments.

We are providing new data comparing the mTORC1 signaling in *Tsc1*^{-/-}, *Tsc2*^{-/-} and WT MEFs (see below Response Fig 4A). Our data shows that total mTOR is reduced in *Tsc1*^{-/-} but not *Tsc2*^{-/-} MEFs. This is because mTOR is a known client of Hsp90. However, this reduction in mTOR protein level does not diminish its activity in the *Tsc1*^{-/-} MEF cells, as shown by pS6K immunoblots. We do not have any mechanistic explanations for this data. We speculate that this is because of differences in Hsp90 chaperone activity between *Tsc1*^{-/-} and *Tsc2*^{-/-} MEFs. Our drug binding assays using the lysates from these two cells lines showed that Hsp90 from *Tsc1*^{-/-} MEFs bound less ganetespib compared to the WT and *Tsc2*^{-/-} MEFs (Fig 4D, Response Fig 4B). We also did not see any changes to drug binding of Hsp90 from WT and *Tsc2*^{-/-} MEFs (Response Fig 4B). This suggests that Hsp90 dynamics are more affected by *Tsc1* vs. *Tsc2* loss, which would have subsequent effects on mTORC1 signaling since many pathway components are clients of Hsp90.

Response Fig 4. mTOR signaling in TSC1 and TSC2 ^{-/-} MEF cells.

A) Lysates from wild-type (WT), *Tsc1* and *Tsc2* ^{-/-} MEF cells were immunoblotted for the components of the mTOR signaling

B) Lysates from HEK293 overexpressing *Tsc2*-FLAG, as well as *Tsc2* ^{-/-} and ^{+/+} MEF cells were incubated with indicated amounts of biotinylated-GB followed by streptavidin agarose-beads. Hsp90 was detected by immunoblotting. Hsp90 and *Tsc2* inputs were presented in the lower panel.

Taken together our data, although preliminary, shows that the Tsc1-Hsp90 connection plays a role in mTORC1 signaling. Because of the complexity of this pathway and requirement of additional data to decipher the mTORC1 signaling, we believe addition of the current data (Response Fig 4) in the manuscript will confuse the readers and eventually dilute our take home message.

With regards to the reviewer's second comment, "*They should also rescue TSC1 null cells with TSC1 constructs that can or cannot bind to Hsp90 and determine their ability to restore mTORC1 regulation*": unfortunately we have not yet identified a Tsc1 mutant that does not interact with Hsp90. Our alternative strategy was to express Tsc1 lacking the D-fragment (Tsc1 aa1-998) in the Tsc1-null cells, therefore disrupting Tsc1 interaction with Hsp90. However, we were concerned about our interpretation of this potential data because we show that Tsc1-D interacts with only the middle domain of Hsp90 (Fig 2E) but deletion of the MEEVD motif (the TPR domain binding site) at the very C-terminus of Hsp90 completely abrogates Tsc1 binding to Hsp90 (Fig 2C). This suggests that the binding of Tsc1 to Hsp90 is complex, and it requires multiple surfaces of Tsc1 to interact with Hsp90. We therefore did not feel that this Tsc1 D-less fragment (Tsc1 aa1-998) was sufficient to answer this comment and would complicate the interpretation of our data.

5- The fact that TSC1 and TSC2 interact with both Hsp90 and Hsp70 raises concerns that the interaction may not be as specific as the authors propose. Since the Hsp70 interaction is shown in the paper, its reproducibility and significance should be documented, at least in part. Based on our findings, Tsc2 is a client protein of both Hsp70 and Hsp90. We also propose that Tsc1 is a new co-chaperone of Hsp90. However, whether Tsc1 functions like the co-chaperone HOP and also regulates Hsp70 (Schmid et al., 2012, Southworth & Agard, 2011), remains unknown. We were careful not to make this speculation in the Discussion, however we have clarified and stated the significance of this finding in our Discussion;

"Like other co-chaperones such as Cdc37 and HOP, Tsc1 behaves as a scaffold in order to load clients, in this case Tsc2, to Hsp90. It is also conceivable that Tsc1 regulates the chaperone function of Hsp70 and perhaps like HOP, connecting the transfer of clients between Hsp70 and Hsp90."

With regards to reproducibility, as we have stated in the Methods section, the data presented in our paper are the representative examples of three biological replicates.

Additional points.

1- The interaction between TSC2 and Hsp90 (Fig 1A) lacks adequate negative controls. IgG is not sufficient, another protein of comparable size to TSC2 should be provided. We thank the reviewer for this comment. Hsp90 is a molecular chaperone that has approximately 200 known client proteins of various molecular weights. Therefore, we do not feel comfortable using a random protein of comparable size to Tsc2 as a negative control in this experiment. The use of IgG as a negative control is a common practice and is generally accepted in the field, as we have used previously in immunoprecipitation experiments of large molecular weight proteins, such as the new co-chaperones FNIP1 and FNIP2 (Woodford et al., 2016a). We have provided additional negative data (please see below and Fig EV1) demonstrating the absence of interaction of Tsc2 with the co-chaperone Aha1. We believe that this will serve as a better negative control and will not confuse the readers, instead of using a random protein of a similar size as Tsc2.

Fig EV1. Tsc2 was immunoprecipitated from HEK293 cell lysates using anti-Tsc2 or IgG (control) and immunoblotted with indicated antibody to confirm the lack of Aha1 co-chaperones interaction with Tsc2.

2- Similarly, in Fig 2A, an adequate negative control must be provided for the interaction of TSC1 with Hsp90.

Similar to point 1, relevant negative controls are already presented in Fig EV2A. These include HOP, p23 and Aha1. Please see above.

Fig EV2A. Tsc2 was immunoprecipitated from HEK293 cell lysates using anti-Tsc1 or IgG (control) and immunoblotted with indicated antibody to confirm the lack of Aha1, p23 and HOP co-chaperones interaction with Tsc1.

3- The ATPase assay in Fig 2G also lacks a negative control protein, therefore it cannot be concluded that TSC1 (D-domain) is a specific Hsp90 inhibitor.

We thank the reviewer for this comment. We have demonstrated the specificity of Tsc1 on Hsp90 ATPase activity using BSA as a negative control (Fig EV2E). We have previously shown that BSA does not affect Hsp90 ATPase activity *in vitro* (Woodford et al., 2016a).

References

- Carpy A, Krug K, Graf S, Koch A, Popic S, Hauf S, Macek B (2014) Absolute proteome and phosphoproteome dynamics during the cell cycle of *Schizosaccharomyces pombe* (Fission Yeast). *Mol Cell Proteomics* 13: 1925-36
- Chong-Kopera H, Inoki K, Li Y, Zhu T, Garcia-Gonzalo FR, Rosa JL, Guan KL (2006) TSC1 stabilizes TSC2 by inhibiting the interaction between TSC2 and the HERC1 ubiquitin ligase. *J Biol Chem* 281: 8313-6
- Crino PB, Nathanson KL, Henske EP (2006) The tuberous sclerosis complex. *The New England journal of medicine* 355: 1345-56
- Deprost D, Yao L, Sormani R, Moreau M, Leterreux G, Nicolai M, Bedu M, Robaglia C, Meyer C (2007) The Arabidopsis TOR kinase links plant growth, yield, stress resistance and mRNA translation. *EMBO reports* 8: 864-70
- Diaz-Troya S, Perez-Perez ME, Florencio FJ, Crespo JL (2008) The role of TOR in autophagy regulation from yeast to plants and mammals. *Autophagy* 4: 851-65
- Dunn DM, Woodford MR, Truman AW, Jensen SM, Schulman J, Caza T, Remillard TC, Loiselle D, Wolfgeher D, Blagg BS, Franco L, Haystead TA, Daturpalli S, Mayer MP, Trepel JB, Morgan RM, Prodromou C, Kron SJ, Panaretou B, Stetler-Stevenson WG et al. (2015) c-Abl Mediated Tyrosine Phosphorylation of Aha1 Activates Its Co-chaperone Function in Cancer Cells. *Cell reports* 12: 1006-18
- Hengstschlager M, Rosner M, Fountoulakis M, Lubec G (2005) The cellular response to ectopic overexpression of the tuberous sclerosis genes, TSC1 and TSC2: a proteomic approach. *Int J Oncol* 27: 831-8
- Hoogeveen-Westerveld M, Wentink M, van den Heuvel D, Mozaffari M, Ekong R, Povey S, den Dunnen JT, Metcalfe K, Vallee S, Krueger S, Bergoffen J, Shashi V, Elmslie F, Kwiatkowski D, Sampson J, Vidales C, Dzarir J, Garcia-Planells J, Dies K, Maat-Kievit A et al. (2011) Functional assessment of variants in the TSC1 and TSC2 genes identified in individuals with Tuberous Sclerosis Complex. *Human mutation* 32: 424-35
- Hornigold N, Devlin J, Davies AM, Aveyard JS, Habuchi T, Knowles MA (1999) Mutation of the 9q34 gene TSC1 in sporadic bladder cancer. *Oncogene* 18: 2657-61
- Kucejova B, Pena-Llopis S, Yamasaki T, Sivanand S, Tran TA, Alexander S, Wolff NC, Lotan Y, Xie XJ, Kabbani W, Kapur P, Brugarolas J (2011) Interplay between pVHL and mTORC1 pathways in clear-cell renal cell carcinoma. *Mol Cancer Res* 9: 1255-65
- Menand B, Desnos T, Nussaume L, Berger F, Bouchez D, Meyer C, Robaglia C (2002) Expression and disruption of the Arabidopsis TOR (target of rapamycin) gene. *Proc Natl Acad Sci U S A* 99: 6422-7
- Miloloza A, Kubista M, Rosner M, Hengstschlager M (2002) Evidence for separable functions of tuberous sclerosis gene products in mammalian cell cycle regulation. *J Neuropathol Exp Neurol* 61: 154-63
- Mollapour M, Bourboulia D, Beebe K, Woodford MR, Polier S, Hoang A, Chelluri R, Li Y, Guo A, Lee MJ, Fotooh-Abadi E, Khan S, Prince T, Miyajima N, Yoshida S, Tsutsumi S, Xu W, Panaretou B, Stetler-Stevenson WG, Bratslavsky G et al. (2014) Asymmetric Hsp90 N Domain SUMOylation Recruits Aha1 and ATP-Competitive Inhibitors. *Mol Cell* 53: 317-29

- Mollapour M, Neckers L (2011) Detecting HSP90 phosphorylation. *Methods Mol Biol* 787: 67-74
- Mollapour M, Tsutsumi S, Donnelly AC, Beebe K, Tokita MJ, Lee MJ, Lee S, Morra G, Bourbouliia D, Scroggins BT, Colombo G, Blagg BS, Panaretou B, Stetler-Stevenson WG, Trepel JB, Piper PW, Prodromou C, Pearl LH, Neckers L (2010) Swe1Wee1-dependent tyrosine phosphorylation of Hsp90 regulates distinct facets of chaperone function. *Mol Cell* 37: 333-43
- Mollapour M, Tsutsumi S, Truman AW, Xu W, Vaughan CK, Beebe K, Konstantinova A, Vourganti S, Panaretou B, Piper PW, Trepel JB, Prodromou C, Pearl LH, Neckers L (2011) Threonine 22 phosphorylation attenuates hsp90 interaction with cochaperones and affects its chaperone activity. *Mol Cell* 41: 672-81
- Neuman NA, Henske EP (2011) Non-canonical functions of the tuberous sclerosis complex-Rheb signalling axis. *EMBO Mol Med* 3: 189-200
- Orlova KA, Crino PB (2010) The tuberous sclerosis complex. *Ann N Y Acad Sci* 1184: 87-105
- Pymar LS, Platt FM, Askham JM, Morrison EE, Knowles MA (2008) Bladder tumour-derived somatic TSC1 missense mutations cause loss of function via distinct mechanisms. *Hum Mol Genet* 17: 2006-17
- Rosner M, Freilinger A, Lubec G, Hengstschlager M (2005) The tuberous sclerosis genes, TSC1 and TSC2, trigger different gene expression responses. *Int J Oncol* 27: 1411-24
- Schmid AB, Lagleder S, Grawert MA, Rohl A, Hagn F, Wandinger SK, Cox MB, Demmer O, Richter K, Groll M, Kessler H, Buchner J (2012) The architecture of functional modules in the Hsp90 co-chaperone Sti1/Hop. *EMBO J* 31: 1506-17
- Scroggins BT, Robzyk K, Wang D, Marcu MG, Tsutsumi S, Beebe K, Cotter RJ, Felts S, Toft D, Karnitz L, Rosen N, Neckers L (2007) An acetylation site in the middle domain of Hsp90 regulates chaperone function. *Mol Cell* 25: 151-9
- Soroka J, Wandinger SK, Mausbacher N, Schreiber T, Richter K, Daub H, Buchner J (2012) Conformational switching of the molecular chaperone Hsp90 via regulated phosphorylation. *Mol Cell* 45: 517-28
- Southworth DR, Agard DA (2011) Client-loading conformation of the Hsp90 molecular chaperone revealed in the cryo-EM structure of the human Hsp90:Hop complex. *Mol Cell* 42: 771-81
- Thien A, Prentzell MT, Holzwarth B, Klasener K, Kuper I, Boehlke C, Sonntag AG, Ruf S, Maerz L, Nitschke R, Grellscheid SN, Reth M, Walz G, Baumeister R, Neumann-Haefelin E, Thedieck K (2015) TSC1 activates TGF-beta-Smad2/3 signaling in growth arrest and epithelial-to-mesenchymal transition. *Dev Cell* 32: 617-30
- van Dam TJ, Zwartkruis FJ, Bos JL, Snel B (2011) Evolution of the TOR pathway. *J Mol Evol* 73: 209-20
- Vernoud V, Horton AC, Yang Z, Nielsen E (2003) Analysis of the small GTPase gene superfamily of Arabidopsis. *Plant Physiol* 131: 1191-208
- Woodford MR, Dunn DM, Blanden AR, Capriotti D, Loiselle D, Prodromou C, Panaretou B, Hughes PF, Smith A, Ackerman W, Haystead TA, Loh SN, Bourbouliia D, Schmidt LS, Marston Linehan W, Bratslavsky G, Mollapour M (2016a) The FNIP co-chaperones decelerate the Hsp90 chaperone cycle and enhance drug binding. *Nature communications* 7: 12037

Woodford MR, Truman AW, Dunn DM, Jensen SM, Cotran R, Bullard R, Abouelleil M, Beebe K, Wolfgeher D, Wierzbicki S, Post DE, Caza T, Tsutsumi S, Panaretou B, Kron SJ, Trepel JB, Landas S, Prodromou C, Shapiro O, Stetler-Stevenson WG et al. (2016b) Mps1 Mediated Phosphorylation of Hsp90 Confers Renal Cell Carcinoma Sensitivity and Selectivity to Hsp90 Inhibitors. *Cell reports* 14: 872-84

Yang Y, Rao R, Shen J, Tang Y, Fiskus W, Nechtman J, Atadja P, Bhalla K (2008) Role of acetylation and extracellular location of heat shock protein 90alpha in tumor cell invasion. *Cancer Res* 68: 4833-42

Zeng LH, Rensing NR, Zhang B, Gutmann DH, Gambello MJ, Wong M (2011) Tsc2 gene inactivation causes a more severe epilepsy phenotype than Tsc1 inactivation in a mouse model of tuberous sclerosis complex. *Hum Mol Genet* 20: 445-54

Thank you for submitting a revised version of your manuscript. It has now been seen by all three of the original referees and their comments are shown below. As you will see, the referees find that the main criticisms have been addressed and they therefore recommend the manuscript for publication in The EMBO Journal, pending minor textual revision. I would therefore like to invite you to submit a final revision of the manuscript in which you discuss the remaining referee concerns and also address the following editorial points:

-> We can accommodate up to 5 keywords per paper and noticed that your manuscript currently has 7, could I ask you to remove two of them?

-> It looks like the same Tsc2 IP is presented for the co-IPs shown in figure 1A and EV1A, could you confirm if this is the case? If so, then please make sure this is mentioned in the figure legend. The same thing goes for the Tsc1 IPs shown in Figure 2A and Fig EV2A

-> Please include the number of replicas used for calculating statistics in the relevant figure legends (currently missing for fig 2G, fig 5J, Fig EV2D-F, Fig EV4B)

-> We generally encourage the publication of source data, particularly for electrophoretic gels and blots, with the aim of making primary data more accessible and transparent to the reader. We would need 1 file per figure (which can be a composite of source data from several panels) in jpg, gif or PDF format, uploaded as "Source data files". The gels should be labelled with the appropriate figure/panel number, and should have molecular weight markers; further annotation would clearly be useful but is not essential. These files will be published online with the article as a supplementary "Source Data". Please let me know if you have any questions about this policy.

Thank you again for giving us the chance to consider your manuscript for The EMBO Journal, I look forward to receiving your final revision.

 REFEREE REPORTS

Referee #1:

In the revised manuscript, the authors have addressed most of my comments. However, in general, the paper remains to be lacking in terms of mechanistic insights.

Also, recently a paper was published that shows the interaction of the TSC complex with the R2TP complex. R2TP is an Hsp90 cochaperone complex. The paper is: Cloutier, P., Poitras, C., Durand, M., Hekmat, O., Fiola-Masson, É., Bouchard, A., Faubert, D., Chabot, B., and Coulombe, B. (2017). R2TP/Prefoldin-like component RUVBL1/RUVBL2 directly interacts with ZNHIT2 to regulate assembly of U5 small nuclear ribonucleoprotein. *Nature Communications* 8, 15615.

The authors need to cite/address this paper.

Referee #2:

This paper demonstrates the mechanistic basis for Tsc1 acting as a Hsp90 cochaperone in the regulation of the client protein Tsc2. Tsc1 acts by antagonizing the function of Aha1, unless Aha1 is phosphorylated. The revised manuscript is clearly improved.

I am wondering, however, how Tsc1 can generally overcome the Aha1 activase function for diverse Hsp90 client proteins, considering the low cellular abundance of Tsc1 (2 orders of magnitude lower than Aha1, 3 orders of magnitude lower than Hsp90 in *S. pombe*). Similar ratios were reported for HeLa cells (Kulak et al, *Nat. Methods* 2014). Also, why should such a crucial factor be present only

in select animals and fungi? Is it not more likely that Tsc1 deletion affects the levels of Hsp90 clients other than Tsc2 by an indirect mechanism, for example via the Tsc2 regulation of the kinase activity of mTORC1?

Referee #3:

In the revised manuscript, Woodford et al have addressed most of the concerns raised by the initial submission and clarified some conceptual points. The mechanism of Hsp90 co-chaperoning by TSC1 is well delineated and supported by the experiments. Interestingly, a similar Hsp90-regulating role for another mTOR regulator, the FNIP1/2 complex, was recently described by the authors, which raises the question of how unique or general each of these mechanisms is. Thus, follow-up studies will be required to fully uncover the physiological roles of the TSC1-Hsp90 interaction.

2nd Revision - authors' response

15 September 2017

Dr Anne Nielsen
Editor
The EMBO Journal
Postfach 1022.40,
69012 Heidelberg,
Germany

UPSTATE
MEDICAL UNIVERSITY

Department of Urology
*Department of Biochemistry &
Molecular Biology*

750 East Adams St
Syracuse, NY 13210
Phone: +1-315-464-8749
Fax +1-315-464-8750
mollapom@upstate.edu
www.mollapourlab.com

September 15, 2017

RE: EMBOJ-2017-96700R

Dear Dr Nielsen,

Thank you for giving us the opportunity to address the reviewer's questions and concerns. We are also grateful to the reviewers for their constructive and insightful comments on our manuscript by Woodford et al., entitled "Tsc1 co-chaperone breaks the asymmetric binding of Aha1 with Hsp90 and inhibits the ATPase activity". We are happy to see that the reviewers agreed to the publication of our work.

Below, we provide a detailed point-by-point response to the reviewers' questions.

Thank you for agreeing to publish our manuscript in *The EMBO Journal*.

Sincerely,

Mehdi Mollapour PhD
Assistant Professor of Urology
Head, Renal Cancer Biology Section
Department of Urology
SUNY Upstate Medical University

We are grateful to the reviewers for recommending our manuscript for publication. Below, we provide a detailed point-by-point response to the reviewers' questions.

Referee #1:

In the revised manuscript, the authors have addressed most of my comments. However, in general, the paper remains to be lacking in terms of mechanistic insights.

Also, recently a paper was published that shows the interaction of the TSC complex with the R2TP complex. R2TP is an Hsp90 cochaperone complex. The paper is: Cloutier, P., Poitras, C., Durand, M., Hekmat, O., Fiola-Masson, É., Bouchard, A., Faubert, D., Chabot, B., and Coulombe, B. (2017). R2TP/Prefoldin-like component RUVBL1/RUVBL2 directly interacts with ZNHIT2 to regulate assembly of U5 small nuclear ribonucleoprotein. *Nature Communications* 8, 15615. The authors need to cite/address this paper.

We thank the reviewer for appreciating our additional data. We respectfully disagree with their comment about our manuscript lacking mechanistic insight.

We would like you to refer to our previous response to this criticism; We have identified Tsc1 as a new co-chaperone and Tsc2 as a new client of Hsp90. Tsc1 decelerates the chaperone cycle by inhibiting Hsp90 ATPase-activity, therefore loading Tsc2 on Hsp90. This prevents Tsc2 from ubiquitination and proteasomal degradation. Our work has also mechanistically demonstrated that Tsc1 carboxy-domain homodimerizes and binds to both protomers of the Hsp90 middle-domain, which is sufficient to inhibit Hsp90 ATPase activity. Tsc1 interacts with the Aha1 binding sites on Hsp90 middle-domain, therefore breaking the asymmetric interaction of Aha1 with Hsp90. Our data has also led us to speculate that Tsc1 prevents the conformational changes in the catalytic loop of Hsp90 and stops the release of R400 from its retracted inactivating conformation consequently inhibiting Hsp90 ATPase activity. The latter statement is only a speculation on how exactly Tsc1 might inhibit Hsp90 ATPase activity, which requires structural data in order to confirm this phenomenon. This is clearly not within the scope of this paper.

It is unfortunate that our response does not satisfy the reviewer's critique; we therefore, respectfully, have to agree to disagree.

We have referenced Cloutier *et al.*, 2017. *Nature Communications* in our Discussion section.

Referee #2:

This paper demonstrates the mechanistic basis for Tsc1 acting as a Hsp90 cochaperone in the regulation of the client protein Tsc2. Tsc1 acts by antagonizing the function of Aha1, unless Aha1 is phosphorylated. The revised manuscript is clearly improved. I am wondering, however, how Tsc1 can generally overcome the Aha1 activase function for diverse Hsp90 client proteins, considering the low cellular abundance of Tsc1 (2 orders of magnitude lower than Aha1, 3 orders of magnitude lower than Hsp90 in *S. pombe*). Similar ratios were reported for HeLa cells (Kulak et al, Nat. Methods 2014)).

We are happy and grateful that the reviewer is pleased with the revised manuscript.

With regards to the reviewer's question, we believe a similar mechanism to Aha1 post-translational modification might be playing a role in increasing the affinity of post-translationally modified Tsc1 binding to Hsp90. We are currently dissecting the PTM of Tsc1 and their impact on binding to Hsp90.

Also, why should such a crucial factor be present only in select animals and fungi?

This is a very interesting observation, however we do not have an explanation for this phenomenon.

Is it not more likely that Tsc1 deletion affects the levels of Hsp90 clients other than Tsc2 by an indirect mechanism, for example via the Tsc2 regulation of the kinase activity of mTORC1?

This is a valid point, however one can make the same argument with the other co-chaperones of Hsp90. In other words it is not the direct effect of the co-chaperones *per se* but the consequential effect on the signaling pathways. However, what has made it difficult for the reviewers to digest our research is the fact that Tsc1 and Tsc2 has always been linked to the mTOR pathway, and therefore any effects of Tsc1 are automatically linked to defects in the mTOR pathway. However, if one looks at our study collectively, the biophysical, biochemical and cell-based assays conclude that Tsc1 is a *bona fide* co-chaperone of Hsp90 and modulation of its activity has detrimental effects not only on the mTOR pathway, but other signaling pathways. We do appreciate that dissecting this in cells is a challenging task.

Referee #3:

In the revised manuscript, Woodford et al have addressed most of the concerns raised by the initial submission and clarified some conceptual points. The mechanism of Hsp90 co-chaperoning by TSC1 is well delineated and supported by the experiments. Interestingly, a similar Hsp90-regulating role for another mTOR regulator, the FNIP1/2 complex, was recently described by the authors, which raises the question of how unique or general each of these mechanisms is. Thus, follow-up studies will be required to fully uncover the physiological roles of the TSC1-Hsp90 interaction.

We would like to thank the reviewer for appreciating our revised manuscript. We are currently conducting further experiments to address the relationship between FNIPs and Tsc1 on Hsp90 function. We are also planning to uncover the physiological role of Tsc1 as a co-chaperone of Hsp90.

Thank you for submitting the final revision of your manuscript, I am pleased to inform you that your study has now been officially accepted for publication in The EMBO Journal.

Corresponding Author Name: Mehdi Mollapour

Manuscript Number: EMBOJ-2017-96700